# SafeFix: Targeted Model Repair via Controlled Image Generation

**Ouyang Xu**                                                          *oxu@utdallas.edu*
*Department of Computer Science*
*The University of Texas at Dallas*

**Baoming Zhang**                                          *baoming.zhang@utdallas.edu*
*Department of Computer Science*
*The University of Texas at Dallas*

**Ruiyu Mao**                                                   *ruiyu.mao@utdallas.edu*
*Department of Computer Science*
*The University of Texas at Dallas*

**Yunhui Guo**                                                 *yunhui.guo@utdallas.edu*
*Department of Computer Science*
*The University of Texas at Dallas*

**Reviewed on OpenReview:** *https://openreview.net/forum?id=TtpW6JiEiW*

## Abstract

Deep learning models for visual recognition often exhibit systematic errors due to under-represented semantic subpopulations. While existing debugging frameworks can identify these failure slices, effectively repairing them remains difficult. Current solutions often rely on manually designed prompts to generate synthetic images—an approach that introduces distribution shift and semantic errors, often resulting in new bugs. To address these issues, we introduce SafeFix, a framework for distribution-consistent model repair via controlled generation that employs a diffusion model to generate semantically faithful images that modify only specific failure attributes while preserving the underlying data distribution. To ensure the reliability of the repair data, we implement a verification mechanism using a large vision–language model (LVLM) to enforce semantic consistency and label preservation. By retraining models on the synthetic data, we significantly reduce errors in rare cases and improve overall performance. Our experiments show that SafeFix achieves superior robustness by maintaining high precision in attribute editing without introducing additional bugs.

## 1 Introduction

Despite strong performance on standard benchmarks, computer vision models often fail on rare semantic subpopulations (Barbu et al., 2019; Gao et al., 2023; Leclerc et al., 2022). Such errors usually arise from dataset bias: some attribute combinations (for example, individuals with red hair color) are rarely represented in the training data (Buolamwini & Gebru, 2018). Finding and fixing these bug slices is important for deploying AI systems in safety-critical applications.

Recent interpretable debugging pipelines can identify failure slices in vision models (Gao et al., 2023; Chen et al., 2023; Singla et al., 2024). For example, HiBug (Chen et al., 2023) uses vision–language models to discover slices that share visual attributes. Once these slices are found, however, repairing them remains difficult. Existing repair strategies mainly follow two directions. Retrieval-based methods (Gao et al., 2023; Singla et al., 2024) collect additional samples from external datasets, but these samples can introduce domain gaps. Generation-based methods, including the repair strategy suggested by HiBug, synthesize new training data with text-to-image models. This avoids dependence on an external image pool, but prompt-based

**Addressing Challenges in Model Repair with Generated Images**

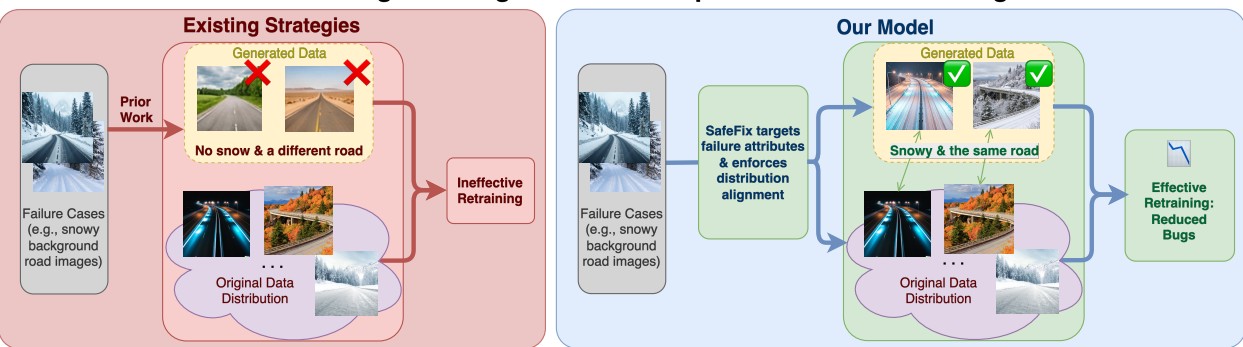

Figure 1: SafeFix addresses model repair challenges by generating images that target specific failure attributes, such as snowy backgrounds. While prior work (Zhang et al., 2024; Chen et al., 2023) may produce data that are insufficiently represented in the source dataset, such methods often miss true failure attributes and cause distribution shifts. In contrast, SafeFix generates images under snowy conditions that maintain the same road geometry to enforce distribution alignment, keeping generated samples consistent with the original data distribution. This approach reduces bugs by improving training coverage of underrepresented semantic subpopulations where model failures stem from insufficient data. It ensures that the augmented dataset effectively repairs the model without introducing new errors.

generation introduces two common failure modes: (1) **Semantic Drift**, where generated images change attributes unrelated to the intended edit, and (2) **Distribution Mismatch**, where synthetic images deviate from the original data distribution and teach the model generative artifacts rather than the intended visual concept.

We propose SafeFix, a targeted repair method that uses controlled image generation to correct these underrepresented slices without degrading overall performance or introducing new errors. Compared with AdaVision (Gao et al., 2023) and DCD (Singla et al., 2024), which retrieve samples from another dataset, SafeFix generates repair examples from the original training distribution. Compared with HiBug (Chen et al., 2023), which follows the same high-level diagnose-generate-retrain structure, SafeFix changes the repair-data construction step. HiBug turns attribute text into a prompt and generates a new image; even when the target attribute appears, source-specific factors such as eye shape, face contour, background, pose, and lighting can change. SafeFix instead conditions generation on real training images, edits the diagnosed attributes, and preserves the other visual factors of the source image.

This distinction is not due to a stronger diffusion backbone; both HiBug and SafeFix use Stable Diffusion 1.5 (Rombach et al., 2022). The latent-space-based filtering method (Jain et al., 2023b) also explores repair via diffusion models, but it relies on caption-based prompting and does not verify the semantic accuracy of critical attributes. SafeFix adds a large vision–language model (LVLM) filter (Bai et al., 2025; Liu et al., 2023), verified by human audit, to remove generated images that miss the intended attribute or change the label. The resulting repair images reflect the intended edit (e.g., a sad expression on a darker-skinned woman with red hair) while staying close to the source image and the original data distribution.

As shown in Figure 1, SafeFix addresses two repair requirements: generated images should contain the diagnosed failure attributes, and they should remain close to the original data distribution. By retraining models on this augmentation set, we reduce errors on underrepresented semantic subpopulations. The contributions of this work are as follows:

- We formulate a targeted model repair pipeline, SafeFix, which leverages conditional text-to-image generation and LVLM-based filtering to synthesize high-quality, attribute-faithful data for correcting model failures arising from underrepresented subpopulations.

- We introduce a verification mechanism that leverages LVLMs to mitigate the inherent unreliability of generative models, ensuring that the synthesized repair data is both semantically accurate and remains aligned with the original dataset distribution.

- We demonstrate that SafeFix improves rare-case failure slices across multiple architectures and datasets, achieving the best performance in the evaluated CelebA, ImageNet10, COCO, and KITTI settings across image classification, pose estimation, and object detection.

## 2    Related Work

**Failure Pattern Discovery.** HiBug (Chen et al., 2023) identifies interpretable failure cases in vision models by clustering semantically meaningful attributes, exposing rare categories and spurious correlations. HiBug2 (Chen et al., 2025) builds on this line with more efficient error-slice discovery and a closed-loop debugging mechanism that improves the coherence and coverage of discovered bugs. Other work studies interpretability for model debugging (Adebayo et al., 2020; 2022), but these methods often rely on fixed failure patterns or lack direct visual grounding. MODE (Vendrow et al., 2023) takes a state-differential view, locating internal model faults and using them to propose data-driven remedies. TCAV (Kim et al., 2018) measures a model's sensitivity to high-level concepts and can correct concept-level spurious activations, while 3DB (Leclerc et al., 2022) builds structured attribute spaces over failure modes to discover underrepresented attributes from visual model errors.

**Targeted Synthetic Augmentation via Diffusion Models.** Diffusion models such as Stable Diffusion (Rombach et al., 2022) and classifier-free guidance (Ho & Salimans, 2022) enable controllable, semantically faithful image synthesis. These models have proven useful in debugging (Casper et al., 2022; Fang et al., 2024; Huang et al., 2024), conditional text-to-image visualization (Augustin et al., 2022; Boreiko et al., 2022), and training data augmentation (Trabucco et al., 2023; Dunlap et al., 2023), especially in few-shot and fine-grained recognition tasks. Recent work like DiGA (Zhang et al., 2024) shows how editing spurious attributes while preserving class semantics can mitigate bias without requiring new annotations. These advances highlight how targeted generation can shape training distributions to address model weaknesses. Building on these insights, our method forms a debugging pipeline that not only identifies failure cases but also synthesizes and integrates targeted images to improve model performance. Compared to prior augmentation or error discovery pipelines (Fang et al., 2024; Huang et al., 2024; Chen et al., 2023), our approach is more generalizable and less reliant on predefined attribute sets. We draw inspiration from efforts like StylizedImageNet (Geirhos et al., 2019), debiasing pipelines (Jin & Rinard, 2021), and adaptive augmentation methods (Mikołajczyk-Bareła et al., 2023; Zhao et al., 2022; Wang et al., 2024b), but focus on semantically controllable generation tailored to discovered bugs.

**Multimodal Filtering via LVLMs.** LVLMs like Flamingo (Alayrac et al., 2022) have demonstrated strong capabilities in semantically grounding visual concepts. Recent studies show that using LVLMs as filters to select high-quality image-text pairs can improve dataset quality for downstream tasks (Wang et al., 2024a; Li et al., 2024). These approaches outperform traditional methods like CLIP-based filtering by providing fine-grained, attribute-aware analysis of generated samples. In line with these trends, we adopt an LVLM as an automated filtering component, grounding our approach in established methods that leverage LVLMs for semantic validation of generated data.

**Targeted Repair for Rare-Case Bugs.** Recent work explores how underrepresented subpopulations induce systematic errors in vision models and how targeted interventions can mitigate such failure modes. DOMINO (Eyuboglu et al., 2022) discovers coherent failure slices by clustering model errors in a cross-modal embedding space, enabling automated identification of rare-case bugs without manual slice definitions. REAL (Parashar et al., 2024) focuses on rare visual concepts that are neglected in large-scale vision–language datasets, augmenting them by retrieving semantically similar examples and fine-tuning lightweight classifiers on the retrieved subsets to improve model robustness on these rare categories. This work reflects a broader shift toward targeted model repair using interpretable diagnostics and subpopulation-aware interventions, aligning with our controlled synthesis and refinement approach.

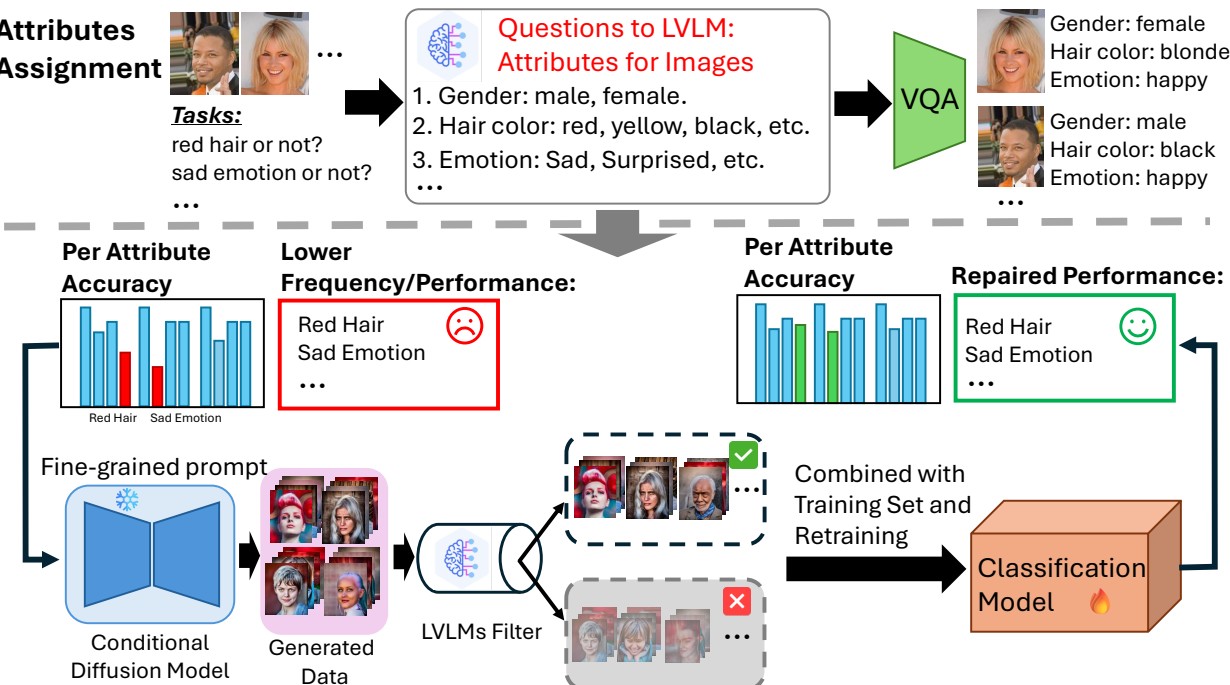

Figure 2: **Overview of SafeFix**. We propose a targeted model repair pipeline that identifies rare-case failures, generates attribute-specific synthetic images using a conditional diffusion model, filters them via a large vision–language model, and retrains the model to improve accuracy and fix rare-case bugs.

## 3 Background

**Attribute-based Model Debugging.** Let $x \in \mathcal{X}$ be an input image with its corresponding ground-truth label $y(x) \in \mathcal{Y}$. A computer vision model $f_\theta \colon \mathcal{X} \to \mathcal{Y}$ produces a prediction $f_\theta(x)$ for each input image $x$. Denote the training, validation, and test splits by $\mathcal{D}_{\text{train}}$, $\mathcal{D}_{\text{val}}$, and $\mathcal{D}_{\text{test}}$, respectively. The full dataset is then given by $\mathcal{D} = \mathcal{D}_{\text{train}} \cup \mathcal{D}_{\text{val}} \cup \mathcal{D}_{\text{test}}$. The overall validation accuracy is computed as:

$$\text{Acc}(\mathcal{D}_{\text{val}}) = \frac{1}{|\mathcal{D}_{\text{val}}|} \sum_{(x,y) \in \mathcal{D}_{\text{val}}} \mathbf{1}\{f_\theta(x) = y(x)\}. \tag{1}$$

In attribute-based model debugging (Chen et al., 2023), each image is assigned several attributes that represent different subpopulations. In particular, let $\mathcal{A} = \{a_1, \ldots, a_m\}$ be a set of attribute functions, where each $a_i \colon \mathcal{X} \to \mathcal{V}_i$ maps an image $x$ to a discrete value $v \in \mathcal{V}_i$ and $\mathcal{V}_i$ represents the set of all possible values for attribute $i$. A slice $S$ is defined as the set of images that satisfy a conjunction of attribute-value assignments:

$$S = \{x \mid a_{j_1}(x) = v_{j_1}, \ldots, a_{j_k}(x) = v_{j_k}\}, \tag{2}$$

where $\mathcal{J} = \{j_1, \ldots, j_k\} \subseteq [m]$ is the index set of attributes involved in the slice $S$. The slice accuracy is

$$\text{Acc}(S) = \frac{1}{|S|} \sum_{x \in S} \mathbf{1}\{f_\theta(x) = y(x)\}. \tag{3}$$

**Underrepresented Semantic Subpopulations.** Rare-case bugs, often caused by underrepresented semantic subpopulations (Eyuboglu et al., 2022), are attribute-based failure modes for which the model's error rate on validation samples matching a target description significantly exceeds its overall error and those samples appear infrequently in the training dataset. We say a slice $S_r$ is a *rare-case slice* if it constitutes less than a fraction $\rho$ of the training data, i.e.,

$$|S_r| < \rho \, |\mathcal{D}_{\text{train}}|, \tag{4}$$

where $\rho$ is a *rare threshold* (e.g., 0.05). We flag a candidate slice $S_r$ as a bug slice $S_e$ if

$$\text{Acc}(S_e) < \text{Acc}(\mathcal{D}_{\text{val}}) - \epsilon. \tag{5}$$

where $\epsilon$ is an *accuracy difference threshold*. Thus, the model shows significantly lower accuracy on the bug slice—which represents an underrepresented semantic subpopulation—relative to its average validation accuracy. A bug slice can be converted to a human-readable bug description (e.g., "people with red hair smiling tend to have low accuracy on the 'wearing lipstick' classification task").

## 4    SafeFix

To address rare-case bugs caused by underrepresented semantic subpopulations, we propose a targeted model repair strategy that leverages controlled image generation and semantic filtering enabled by a large vision–language model. Our goal is to generate synthetic images that accurately represent underrepresented semantic subpopulations while ensuring these images remain aligned with the training distribution. The overall workflow is summarized in Figure 2.

We begin by generating visually controlled images using a text-to-image diffusion model, *Stable Diffusion*, guided by a structure-conditioned controller, ControlNet (Zhang et al., 2023). Instead of relying solely on language prompts, we generate images conditioned on training data while modifying specific attributes to reflect failure slice semantics. Next, we employ an LVLM to automatically verify whether each generated image accurately reflects the intended attributes. This filtering step ensures that only semantically faithful samples are retained. Finally, we augment the original training dataset with the validated images and retrain the model. SafeFix enhances performance on error-prone regions without compromising overall accuracy or introducing new bugs.

### 4.1    Model Diagnosis for Identifying Rare-Case Bugs

We begin by training a standard computer vision model $f_\theta$ on the original training set $\mathcal{D}_{\text{train}}$. After obtaining predictions on the validation set $\mathcal{D}_{\text{val}}$, similar to HiBug, we use a large vision–language model (LVLM; e.g., GPT-4 with vision (OpenAI, 2023)) to propose candidate attributes and a VQA model (e.g., BLIP (Li et al., 2022)) to assign attribute values $a_i(x)$ to each image. We extract a set of rare-case bug slices $\{S_e\}$, each defined by a conjunction of attribute conditions that exhibit both high error rates and low coverage in the dataset. Specifically, to identify underrepresented and error-prone subpopulations, we analyze each attribute $a_i$ and its associated values $v \in \mathcal{V}_i$. For each value $v$, we examine the slice $S = \{x \mid a_i(x) = v\}$ and compute two quantities: (1) its proportion in the training set, and (2) the model's accuracy on the corresponding validation samples.

We flag the slice $S$ as a rare-case bug if:

- The training support $|S \cap \mathcal{D}_{\text{train}}|$ is below the threshold $\rho \cdot |\mathcal{D}_{\text{train}}|$.

- The validation accuracy $\text{Acc}(S)$ is significantly below the overall validation accuracy $\text{Acc}(\mathcal{D}_{\text{val}})$ (by a margin $\epsilon$).

For example, consider the attribute *hair color*. Suppose only 3% of the dataset have *red hair*, and the model performs poorly on this group (e.g., 60% accuracy versus 85% overall). This makes *red hair* a rare-case bug. In contrast, if *yellow hair* is infrequent but achieves high accuracy, it is not considered a bug. This analysis helps isolate specific attribute values that contribute to systematic model failures.

### 4.2    Targeted Generation with Conditional Diffusion Models (CDMs)

Next, we aim to produce attribute-preserving edits on each original image $x$, focusing on targeted attributes identified in problematic slices from the previous diagnostic stage. Specifically, we generate visually controlled synthetic images $x'$ using a text-to-image diffusion model, *Stable Diffusion*, guided by the structure-conditioned controller ControlNet (Zhang et al., 2023).

To determine which attribute–value pairs to edit, we first identify rare-case slices $S_e$ by computing slice support and validation accuracy across attribute conjunctions, as defined in Eq. equation 4 and Eq. equation 5. Each $S_e$ contains one or more attribute–value pairs $\{(a_j, v'_j)\}_{j \in \mathcal{J}}$ that are both infrequent and underperforming. These attributes yield significantly lower accuracy than average on the validation set. We rank all such attributes by validation error and select the top-$k$ attributes for augmentation. For each original image $x$ that does not satisfy all conditions in $S_e$, we construct an edited variant $x'$ by modifying the selected attributes $\{a_j\}_{j \in \mathcal{J}}$ to match the error-prone configuration defined by $S_e$, while preserving all other visual characteristics and keeping the original label unchanged, i.e., $y(x') = y(x)$.

Given an original image $x$ in the training set with attribute assignment $\{(a_i(x) = v_i)\}_{i=1}^m$, we construct a modified image $x'$ by replacing a subset $\{(a_j, v_j)\}_{j \in \mathcal{J}}$ with $\{(a_j, v'_j)\}_{j \in \mathcal{J}}$, where $\mathcal{J} \subseteq [m]$ indexes attributes satisfying the slice condition $S_e$. In practice, this subset is small (i.e., $|\mathcal{J}| \ll m$), and the remaining attribute assignments $\{(a_i, v_i)\}_{i \in [m] \setminus \mathcal{J}}$ are left unchanged. For example, suppose $x$ has attribute assignment (*black hair*, *happy emotion*) and label "not wearing lipstick." If both attributes are part of the rare-case slice $S_e$, then the synthetic variant $x'$ is generated with attributes (*red hair*, *sad emotion*) while retaining the same label "not wearing lipstick." This attribute-preserving image generation aligns with fairness-driven augmentation in attribute classification, where rare attributes (e.g., hair color or emotion) are modified while the primary label is held fixed. By retraining the model on these synthetically augmented samples $\{x'\}$, we encourage the model to correct rare-case bugs without introducing new bugs.

### 4.3 Filtering via Large Vision–Language Models

We found that synthetic images generated by the Conditional Diffusion Model (CDM) can sometimes fail to accurately reflect the intended attribute modifications. To address this, we employ large vision–language models (LVLMs), Qwen2.5-VL-7B (Bai et al., 2025) and LLaVA-v1.5-7B (Liu et al., 2023), to automatically verify that each generated image correctly exhibits the desired attributes and retains the original label.

Specifically, for each synthetic image $x'$ generated to satisfy a bug slice $S_e$, which contains the error attribute-value pairs responsible for bugs, we iterate over each edited pair $(a_j, v'_j)$ for $j = 1, \ldots, k$, and query the LVLM with:

```
"Does the object have attribute a_j equal to v'_j?"
       "Is the object in this picture labeled y(x)?"
```

We retain only those images for which the LVLM answers "yes" to all queries and confirms that the label matches the original ground truth $y(x)$. Thus, after the LVLM filtering, the generated image $x'$ satisfies the desired attributes and preserves the original label, i.e., $y(x') = y(x)$. These validated samples are then added to $\mathcal{D}_{\text{train}}$ for model retraining. We conduct human-audit experiments to verify that the LVLMs can reliably filter out low-quality generated samples.

### 4.4 Combining Generated Images with the Original Dataset and Retraining

To repair the rare-case bugs while maintaining the model's performance on the overall training distribution (Lee et al., 2024), we augment the training set by adding the validated synthetic images $\{x'\}$, yielding an updated training set

$$\mathcal{D}'_{\text{train}} = \mathcal{D}_{\text{train}} \cup \{x'\}.$$

We then retrain the vision model $f_\theta$ on $\mathcal{D}'_{\text{train}}$ and evaluate its performance on $\mathcal{D}_{\text{val}}$. We report both the overall accuracy improvement and the reduction in failure rates (fix rate) on critical rare-attribute slices $S_e$, before and after augmentation.

## 5 Results

### 5.1 Experimental Setup

**Datasets.** We evaluate our method on two classification tasks that exhibit attribute-based failure modes and we use an 8:1:1 split for training, validation, and testing, respectively.

*Lipstick-wearing classification.* We use the CelebA dataset (Liu et al., 2015) and follow the same data split protocol as (Chen et al., 2023) (80,000, 10,000, 10,000 for train/val/test). The task is to predict whether a person is wearing lipstick, a label known to be correlated with other attributes such as gender and hair color. In the following experiments, we refer to this dataset simply as *CelebA*.

*ImageNet-10 classification.* We construct a 10-class subset of ImageNet (Deng et al., 2009) containing the following categories: `backpack`, `barber chair`, `coffee mug`, `desk`, `electric guitar`, `park bench`, `pitcher`, `purse`, `rocking chair`, and `water bottle`. Each class contains 1,300 images. This subset is selected to study classification failures related to visual attributes such as texture and color. In the following experiments, we refer to this dataset simply as *ImageNet10*.

**Baselines.** We compare our method with six recent baselines that are either data augmentation or use generative augmentation strategies:

- **Data Augmentation**. We apply on-the-fly augmentations to each training image, including random resizing, flipping, color jittering, grayscale conversion, erasing, and normalization, while keeping the dataset size fixed. At test time, images are resized, center-cropped, and normalized deterministically.

- **DiGA (Zhang et al., 2024)**. This method utilizes a two-stage framework to automatically detect spurious attributes and modify them with varying degrees of intensity. It keeps the target attribute constant while diversifying other features to mitigate the effect of spurious correlations on model performance.

- **DA-CDM (Fang et al., 2024)** is a data augmentation method for object detection. It uses a controllable diffusion model guided by visual priors from original images, which enables direct reuse of existing bounding box annotations. It then applies a category-calibrated CLIP score to filter generated data and ensure high-quality, text-aligned samples.

- **Mask-ControlNet (Huang et al., 2024)** is a text-guided image generation pipeline that uses ControlNet with facial occlusion masks to synthesize diverse face images under specific occlusions as a data augmentation method to improve model robustness.

- **HiBug_Class (Chen et al., 2023).** A Class-level method that augments training data with synthetic images. For each class, it uses a diffusion model with a prompt:

  *"A photo of (\*label)."*

- **HiBug_Task (Chen et al., 2023).** A Task-level variant of HiBug_Class that targets failure-prone attributes. It selects attribute slices with the highest validation error and generates prompts to guide a diffusion model:

  *- CelebA*: *"A photo of a {gender} {beard clause} {makeup clause} {lipstick clause} (\*label), with {hair} hair and {skin} skin, looking {emotion}, appearing {age}."*

  *- ImageNet10*: *"A photo of a {color} {class name} (\*label) with {texture} texture, located {object position}, appearing {object size}, in a {background}, under {lighting} lighting, during {time}, from a {perspective} perspective."*

  *Note:* Unless otherwise specified, "HiBug" refers to this optimized "HiBug_Task" variant in the paper. We do not compare with HiBug2 (Chen et al., 2025), which is a data selection method and differs from the synthetic generation strategy. HiBug is the closest generation baseline: it generates new images from attribute prompts, whereas SafeFix edits real training images and verifies the edited outputs with LVLMs. A filtered HiBug image can contain the requested attribute but still lose source-specific visual factors, while SafeFix keeps the repair image anchored to the original example.

**Metrics.** We focus on **targeted improvements for rare slices**, which are critical for fairness and safety, while maximizing overall classification accuracy. Following the standard practice of reporting relative error reduction in machine learning classification evaluation (Manning & Schutze, 1999), we report ***Relative Error Reduction*** (**RER**), defined as

Table 1: RER (%) on CelebA for varying numbers of added images, models, and methods. Ours (L) and Ours (Q) denote LLaVA-7B and Qwen-7B as the large vision–language model filters, respectively. The highest RER in each column is marked in bold.

| Method | ResNet (base acc: 90.57%) | | | ViT (base acc: 85.02%) | | | CLIP (base acc: 88.32%) | | |
|---|---|---|---|---|---|---|---|---|---|
| | 1k Images | 5k Images | 10k Images | 1k Images | 5k Images | 10k Images | 1k Images | 5k Images | 10k Images |
| Data Augmentation | 2.45 | 1.82 | 3.14 | 4.27 | 2.56 | 3.89 | 1.12 | 4.73 | 2.31 |
| DiGA | 2.34 | 4.18 | 3.57 | 5.91 | 4.82 | 5.16 | 3.44 | 2.89 | 6.03 |
| DA-CDM | 4.03 | 3.29 | 3.08 | 4.07 | 10.21 | 6.81 | 15.32 | 16.52 | 15.58 |
| Mask-ControlNet | 5.41 | 7.32 | 6.15 | 11.28 | 10.48 | 12.88 | 16.10 | 15.92 | 16.78 |
| HiBug_Class | 5.09 | 5.83 | 3.40 | 5.14 | 7.74 | 7.21 | 14.64 | 14.98 | 13.96 |
| HiBug_Task | 6.79 | 4.88 | 3.92 | 18.69 | 11.75 | 7.74 | 15.75 | 15.15 | 14.64 |
| **Ours (L)** | 10.39 | 11.45 | 11.66 | 18.09 | **18.42** | 15.35 | 22.26 | 22.52 | **21.75** |
| **Ours (Q)** | **14.32** | **12.09** | **14.95** | **19.29** | 15.42 | **15.95** | **22.77** | **23.12** | 20.46 |

Table 2: RER (%) on ImageNet10 under varying image counts, models, and methods.

| Method | ResNet (base acc: 71.73%) | | | ViT (base acc: 97.42%) | | | CLIP (base acc: 93.78%) | | |
|---|---|---|---|---|---|---|---|---|---|
| | 100 Images | 500 Images | 1k Images | 100 Images | 500 Images | 1k Images | 100 Images | 500 Images | 1k Images |
| Data Augmentation | 3.52 | 2.19 | 4.67 | 1.05 | 3.44 | 2.78 | 4.12 | 1.56 | 3.93 |
| DiGA | 2.15 | 5.67 | 3.98 | 4.41 | 3.22 | 6.12 | 5.49 | 4.33 | 2.76 |
| DA-CDM | 5.24 | 7.32 | 8.53 | 6.20 | 13.18 | 13.57 | 2.89 | 4.34 | 5.95 |
| Mask-ControlNet | 1.73 | 2.87 | 2.33 | -6.59 | 10.47 | 3.88 | -5.47 | 1.61 | -1.29 |
| HiBug_Class | 1.80 | 1.49 | 0.74 | -11.63 | -13.57 | -5.43 | 2.25 | 6.91 | 0.80 |
| HiBug_Task | 6.30 | 4.07 | 5.77 | 9.30 | 13.95 | 6.20 | -7.88 | 6.43 | 5.95 |
| **Ours (L)** | 8.38 | **8.70** | 7.75 | 30.62 | **29.07** | 26.74 | 11.41 | 16.40 | 10.61 |
| **Ours (Q)** | **9.13** | 6.01 | **11.14** | **31.01** | 25.97 | **38.37** | **12.70** | **17.68** | **18.81** |

$$\text{RER} = \frac{\text{Err}_{\text{before}} - \text{Err}_{\text{after}}}{\text{Err}_{\text{before}}} = \frac{(1 - Acc_{\text{before}}) - (1 - Acc_{\text{after}})}{1 - Acc_{\text{before}}} = \frac{Acc_{\text{after}} - Acc_{\text{before}}}{1 - Acc_{\text{before}}}, \quad (6)$$

where $\text{Err} = 1 - Acc$. This metric measures the fraction of the original aggregate error rate removed by repair. Here, $Acc_{\text{after}}$ denotes the accuracy obtained after applying the proposed method, and $Acc_{\text{before}}$ represents the baseline accuracy from standard training using the vision model.

**Implementation Details.** All experiments use an NVIDIA A100 GPU. We take the CelebA "wearing lipstick" classification task as an example. We evaluate three backbone architectures for this classification task: ResNet-18 (He et al., 2016), ViT-B/16 (Dosovitskiy et al., 2020), and CLIP (ViT-B/32) (Radford et al., 2021), all of which are initialized with random weights. This design allows us to isolate the effectiveness of SafeFix from pre-existing biases inherent in pre-trained weights, such as the latent knowledge of ImageNet. Moreover, retraining on the original large-scale pre-training datasets is often infeasible as they may be unknown or inaccessible. All vision models are trained using cross-entropy loss. For rare-case bug discovery, we set the rarity threshold $\rho = 0.05$ and the accuracy difference threshold $\epsilon = 0.03$. For synthetic augmentation, we use ControlNet (Zhang et al., 2023), a CDM based on Stable Diffusion 1.5 (Rombach et al., 2022) and conditioned on soft HED boundaries (Xie & Tu, 2015), with 30 DDIM inference steps. Soft HED boundaries preserve structural details, making this approach suitable for attribute-preserving edits like recoloring and stylizing. To filter generated images, we employ the large vision–language models (LVLMs) Qwen2.5-VL-7B (Bai et al., 2025) and LLaVA-v1.5-7B (Liu et al., 2023). Generating 1,000 images with ControlNet takes about one hour, and filtering these 1,000 images with the LVLM takes ten minutes. Filtering accuracy for most attributes (e.g., hair color, skin tone) exceeds 90%, which is consistent with the human audit in Section 5.7. Combining three attributes yields at least a 70% pass rate for filtered images, showing that the diffusion model produces high-quality samples and that the LVLM filtering is reliable.

## 5.2 Main Results

We summarize the main results on the CelebA and ImageNet10 datasets in Tables 1 and 2, respectively.

On **CelebA**, we select rare-case bugs defined by attribute–value combinations `red hair`, `brown skin`, and `sad emotion` for ResNet and ViT, and `yellow hair`, `brown skin`, and `sad emotion` for CLIP, based on the most frequent patterns identified among failure slices. SafeFix consistently achieves the highest test accuracy, i.e., the highest RER, across all models (ResNet, ViT, and CLIP) and varying levels of synthetic augmentation. For example, with 1,000 added images, our method achieves RER values of 14.32% (ResNet), 10.29% (ViT), and 22.77% (CLIP) relative to their base accuracies. These results show that our attribute-targeted augmentation and filtering pipeline is effective in repairing rare-case failure slices, outperforming both CDM-based and HiBug baselines.

On **ImageNet10**, similar trends emerge, as shown in Table 2. For ResNet and ViT, we target rare-case bugs involving `pink color` and `fabric texture`, while for CLIP we use `orange color` and `fabric texture`. Across all models, our proposed method consistently surpasses the baseline methods. Specifically, our method achieves an RER of 11.14% for ResNet with 1,000 images. ViT and CLIP also exhibit a steady improvement compared to other methods. Table 3 shows that the proposed SafeFix also achieves better overall accuracy on **ImageNet10** compared with the baselines.

**Analysis.** Combined with Tables 1, 2, 3, and *all other test-accuracy results* in Appendix F, our experiments show that baseline performance is often unstable. For instance, baselines like DA-CDM and Mask-ControlNet show some improvement on object and facial attribute tasks, respectively, due to their use of conditional diffusion models. However, they are **not targeted model repair methods**, so their overall performance is inferior to both HiBug and SafeFix. This lack of a targeted strategy means their gains are task-specific and **not robustly transferable**. Similarly, the accuracy of HiBug does not improve substantially. The main limitation is prompt-only generation: even when a generated image contains the requested attribute, it is not an edit of a specific source image and can change non-target factors

Table 3: Test accuracy (%) on ImageNet10 using ResNet-18.

| Method | ResNet | | |
|---|---|---|---|
| | 100 Images | 500 Images | 1k Images |
| Base | | 71.73 | |
| Data Augmentation | 72.73 | 72.35 | 73.05 |
| DiGA | 72.34 | 73.33 | 72.86 |
| DA-CDM | 73.21 | 73.80 | 74.14 |
| Mask-ControlNet | 72.22 | 72.54 | 72.39 |
| HiBug_Class | 72.24 | 72.15 | 71.94 |
| HiBug_Task | 73.51 | 72.88 | 73.36 |
| **Ours (L)** | 74.10 | **74.19** | 73.92 |
| **Ours (Q)** | **74.31** | 73.43 | **74.88** |

such as face shape, background, pose, and lighting. As shown in Figure 3, adding LVLM filtering can reject visibly invalid samples, but filtering alone cannot enforce consistency with a particular original training image. SafeFix succeeds because the LVLM operates on candidates that are already tied to the source distribution through conditional image editing.

In contrast, **our method (SafeFix) consistently shows stable or improving performance**. This robustness indicates our attribute-targeted augmentation and LVLM filtering are highly effective at correcting failure-prone subpopulations with meaningful data.

## 5.3 Extension to Pose Estimation and Object Detection

To test whether SafeFix extends beyond the two classification settings, we also evaluate it on denser visual prediction tasks with larger training sets. For pose estimation, we follow HiBug2 (Chen et al., 2025) on COCO person keypoints (Lin et al., 2014) using Keypoint R-CNN (He et al., 2017) with a ResNet-50-FPN backbone, 20,000 source training images, and 1,173-image held-out validation and test splits, and we report test AP. For object detection, we use KITTI (Geiger et al., 2012) Car and Pedestrian instances converted to YOLO format with 5,000/1,240/1,241 train/validation/test images, train YOLOv8n (Jocher et al., 2023), and report test mAP50-95. As shown in Table 4, SafeFix improves over both the base model and the strongest competing baseline on both tasks. The strongest baseline is HiBug_Task in both settings.

These results do not replace a full ImageNet-scale pretrained-classifier study, but they show that the same diagnose–generate–filter–retrain pipeline works on larger datasets and denser prediction tasks, including settings with standard pretrained visual backbones.

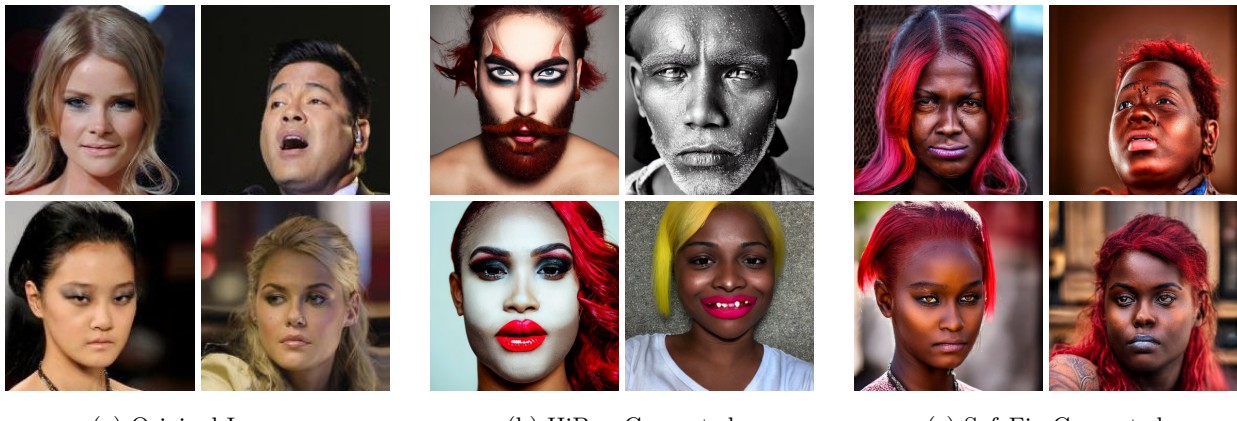

(a) Original Images      (b) HiBug-Generated      (c) SafeFix-Generated

Figure 3: Comparison of generated images from different methods with edited attributes `red hair`, `brown skin`, and `sad emotion`. HiBug often produces invalid or imprecise samples due to the lack of conditional generation and semantic filtering. In contrast, SafeFix generates attribute-faithful images that specifically target rare-case bugs.

Table 4: Results on denser visual prediction tasks. Pose estimation reports COCO keypoint AP, and object detection reports mAP50-95.

| Task | Dataset / Model | Base | HiBug_Task | SafeFix |
|------|----------------|------|-----------|---------|
| Pose estimation | COCO / Keypoint R-CNN R50-FPN | 0.5902 | 0.6145 | **0.6280** |
| Object detection | KITTI / YOLOv8n | 0.5626 | 0.5713 | **0.5790** |

### 5.4 SafeFix Can Effectively Fix Rare-case Bugs

To verify that SafeFix's improvements specifically address targeted rare-case bugs rather than merely enhancing overall performance, we analyze attribute-level validation accuracy changes on CelebA and ImageNet10, as shown in Figure 4. For clarity, the ImageNet10 plot includes only three representative attributes—`color`, `background`, and `texture`—as the dataset contains many attribute dimensions. Specifically, the left plot highlights improvements for CelebA after adding 5,000 synthetic images to the original 80,000 training samples. The right plot demonstrates accuracy gains on ImageNet10, achieved by adding 100 synthetic images to the original training set of 10,400 samples.

Taking ImageNet10 as an example, our targeted synthetic augmentation on `pink color` and `fabric texture` significantly improved accuracy for these selected rare-case attributes. Accuracy for the `pink color` attribute increased from 69.90% to 74.76%, surpassing the overall accuracy of 74.31%. Similarly, accuracy for the `fabric texture` attribute improved from 65.85% to 75.61%, also exceeding the overall accuracy. In contrast, attributes not explicitly targeted by augmentation, such as the `rocks` background, exhibited minimal or no improvement—its accuracy remained unchanged—highlighting that the gains from SafeFix are concentrated on the intended rare-case bugs rather than uniformly distributed across all attributes.

These substantial attribute-specific improvements confirm that SafeFix effectively repairs identified rare-case failure slices rather than providing a generalized performance boost. SafeFix also shows that all attributes improve across both datasets and introduces **no new bugs**, which indicates that the method remains stable outside the targeted regions. Attributes not targeted by augmentation show negligible accuracy changes, further reinforcing that SafeFix precisely and safely addresses the **targeted** rare-case failures.

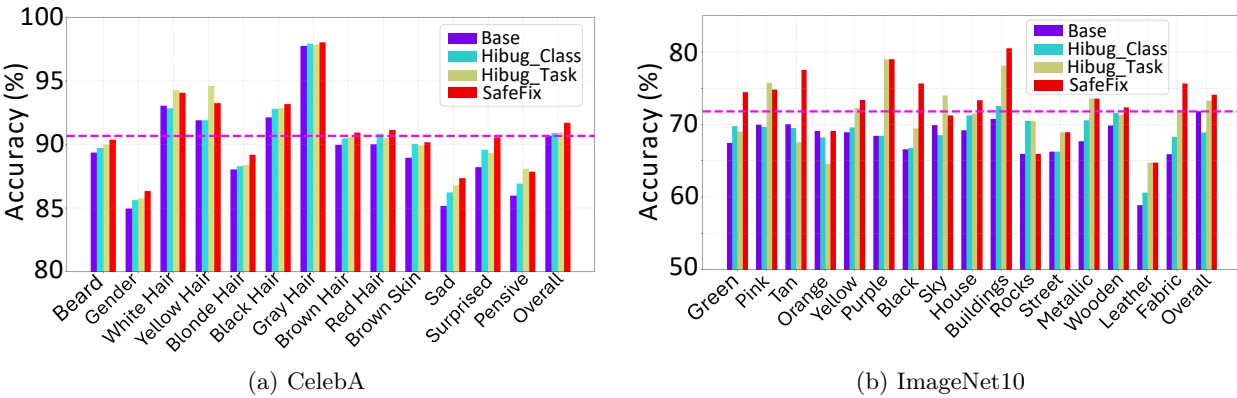

(a) CelebA          (b) ImageNet10

Figure 4: Accuracy comparison of ResNet-18 trained using different augmentation methods. The dashed line represents the average overall accuracy without additional synthetic training data.

### 5.5 SafeFix Directly Addresses Diagnosed Failure Modes, Not Merely Augments Data

To verify that SafeFix's performance gains stem from accurately fixing rare-case bugs rather than from generic data augmentation, we compare red-hair and yellow-hair augmentations on CLIP. As discussed in Section 5.2, we select red hair for ResNet and yellow hair for CLIP in the CelebA dataset, based on which attribute is more likely to trigger rare-case bugs. For CLIP, both `red hair` and `yellow hair` are low-frequency attributes that meet the rarity criterion in Eq. equation 4. However, only the `yellow hair` slice additionally satisfies the low-accuracy criterion in Eq. equation 5, making it a true rare-case bug slice for CLIP.

Figure 5 confirms that augmenting 1,000 images using yellow-hair samples—the diagnosed failure mode for CLIP—substantially improves accuracy on its target slice (**+4.51%**). Furthermore, this targeted augmentation provides positive gains across all other attributes, including on the "Red Hair" slice (+0.91%) and a significant boost to **"Overall" accuracy (+2.66%)**. Conversely, augmenting with red hair, which is not a diagnosed bug, reveals a harmful outcome. Most notably, it degrades performance on its own target slice by **-0.79%**. This counter-intuitive result contrasts with the positive gains seen for ResNet (Figure 4a), highlighting that different architectures can react to synthetic data in unpredictable ways. While red-hair augmentation does provide minor gains

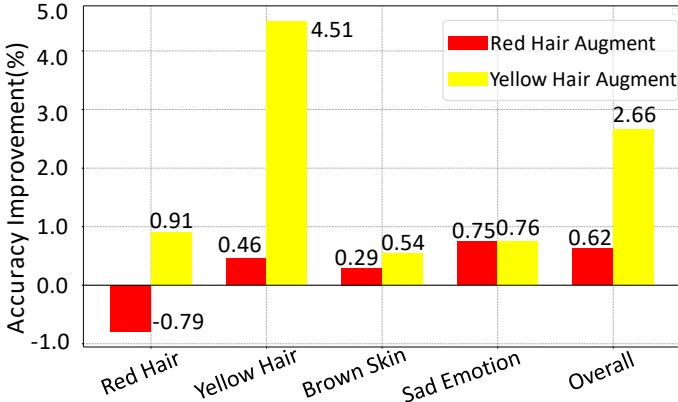

Figure 5: Test accuracy (%) improvements for red-hair vs. yellow-hair augmentation on CLIP.

on other attributes (e.g., +0.75% on `sad emotion`), its minimal overall accuracy impact (+0.62%) and significant negative side effect on the targeted class itself underscore the importance of our diagnosis-driven approach. **Augmenting the *correctly diagnosed* bug (`yellow hair`) leads to effective and safe repair**, while augmenting an *undiagnosed* attribute (`red hair`) can be ineffective and harmful.

### 5.6 Effect of Alternative Bug Slice Selections

Table 5 reports CelebA RER for ResNet after augmenting the training set with different subsets of rare-case attributes. **RH**, **BS**, and **SE** refer to `red hair`, `brown skin`, and `sad emotion`, respectively. Combined settings like **RH_BS** and **RH_BS_SE** (Ours) augment images with multiple attributes. Augmentation improves over the base model (90.57%) across all settings. Notably, our three-way combination, **RH_BS_SE**,

Table 5: RER (%) for attribute-variant selections. Ours is the RH_BS_SE combination.

| # images | RH | BS | SE | RH_BS | RH_SE | RH_BS_SE |
|---|---|---|---|---|---|---|
| 1,000 | 7.85 | 8.59 | 11.77 | 7.74 | 10.39 | **14.32** |
| 5,000 | 10.82 | 12.62 | 11.77 | **14.42** | 6.57 | 12.69 |
| 10,000 | 11.03 | 10.29 | 12.62 | 9.97 | 9.33 | **14.95** |

achieves the highest RER at $1,000$ and $10,000$ images and remains competitive at $5,000$ images, showing that targeting intersecting rare-case conditions is more effective than augmenting isolated attributes alone. Although **RH_BS** alone achieves the highest RER at $5,000$ images, augmenting only that slice does not address concurrent failure modes, indicating persistent errors. First, our **RH_BS_SE** combination improves all three slices concurrently (Figure 4a). Second, for the **RH_BS_SE** attribute combination, the number of bugs decreases from 10 to 4, while augmenting only **SE** or **RH_BS** reduces the count to 7 and 6, respectively (using $\rho = 0.05$ and $\epsilon = 0.03$). A similar pattern holds on ImageNet10: selecting the `pink color + fabric texture` slice reduces rare-case bugs from 13 to 7, whereas augmenting only `pink color` reduces them to 9 and only `fabric texture` reduces them to 10.

After further verification, this reduction does not introduce any new bugs that were not present before, indicating that SafeFix repairs existing failures without adding new bugs on both datasets. The number of identified bugs is sensitive to these thresholds and using different threshold values can change the number of bugs fixed, as shown in Appendix H.7. We note that larger attribute combinations (4 to 7 attributes) also operate correctly but reach lower RER (around 9% on CelebA and 7% on ImageNet10), falling below our selected attribute groups.

### 5.7 Effect of the LVLM Filter and Human Audit

We use an LVLM as the attribute filter, and a human-based test yields nearly identical results. In a human audit conducted with 5 AI graduate students on 300 CelebA images, the pass rates are 97% for "red hair", 99% for "brown skin", 78% for "sad emotion", and 98% for the original label, with an average overlap of about 95% with Qwen2.5-VL (Bai et al., 2025) and 93% with LLaVA-v1.5 (Liu et al., 2023), showing that the LVLMs closely match human verification. The main errors arise from background color changes (**BG**) (7%) and the edited image failing to express the target attribute (**ATTR**) (24%); the LVLM fixes most of them, as shown in Table 6.

Table 6: Average error rates (%) before and after LVLM filtering.

| Error Type | Before | After |
|---|---|---|
| BG | 7 | 2 |
| ATTR | 24 | 3 |

### 5.8 Ablation Study

We ablate the contributions of the two core components in our pipeline: the Conditional Diffusion Model (CDM) and the LVLM filter. Table 7 reports RER on CelebA when using different combinations of these components across three augmentation scales. Configuration (a) corresponds to the prompt-only baseline strategy from HiBug (Chen et al., 2023). Configuration (b) adds LVLM filtering to HiBug-style prompt generations. The filter can remove images that miss the requested attribute, but it cannot make a prompt-generated image preserve the eye shape, face contour,

Table 7: Ablation on the impact of CDM and LVLM components at different scales, reported as RER (%).

| Components | | CelebA RER (%) | | |
|---|---|---|---|---|
| CDM | LVLM | 1,000 | 5,000 | 10,000 |
| (a) | | 6.79 | 4.88 | 3.92 |
| (b) | ✓ | 7.21 | 4.77 | 6.99 |
| (c) | ✓ | | 8.06 | 8.27 | 10.07 |
| (d) | ✓ | ✓ | **14.32** | **12.09** | **14.95** |

background, pose, lighting, and other non-target factors of a specific source training image. Configuration (c) uses source-image-conditioned editing without LVLM verification, which preserves more non-target visual factors but can retain failed edits. SafeFix combines both components in configuration (d) and yields the highest performance, showing that conditional image editing and semantic filtering address different failure modes.

## 6 Conclusion

We presented an automated model repair pipeline for vision tasks that combines failure-attribute diagnostics with targeted synthetic augmentation. We use a Conditional Diffusion Model (CDM) to generate attribute-preserving variants and apply LVLM-based filtering to ensure semantic correctness, thereby focusing augmentation on true rare-case failure slices. Experiments on CelebA and ImageNet10 with ResNet, ViT, and CLIP backbones show consistent gains in accuracy and reduced bugs on underrepresented subpopulations, outperforming CDM-based and HiBug baselines. Additional results on COCO person keypoints and KITTI object detection show that the same repair pipeline extends to denser visual prediction tasks. Ablations confirm that the CDM and the LVLM contribute complementary benefits, highlighting the importance of targeted, validated augmentation for robust model repair.

**Limitations.** Our method's effectiveness is constrained by its components. Crucially, the pipeline can inherit biases from the diffusion model or the LVLM, potentially perpetuating fairness issues if these components have demographic bias. An LVLM filter may reject some demographic groups at different rates; in such cases, the pipeline can generate a larger initial candidate pool to reach the target number of accepted samples, but this increases compute and does not remove the need for better calibrated or debiased verifiers. Other limitations include the computational cost of generation and filtering, potential image artifacts, and a fixed attribute vocabulary that cannot address unmodeled failure modes. Future work will focus on mitigating inherited biases, improving efficiency, adaptive filtering thresholds, and dynamic attribute discovery.

## 7 Acknowledgement

We would like to thank the anonymous reviewers for their helpful comments. This project was partially funded by The University of Texas at Dallas Office of Research and Innovation through the SPIRe grant program. This research was also partially supported by the National Science Foundation (NSF) under Grant No. 2513070.

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

# A    Algorithm

---

**Algorithm 1** SafeFix: Targeted Model Repair via Controlled Image Generation

---

**Require:** Training set $\mathcal{D}_{\text{train}}$, validation set $\mathcal{D}_{\text{val}}$, attribute set $\mathcal{A} = \{a_1, \ldots, a_m\}$, rarity threshold $\rho$, accuracy gap threshold $\epsilon$, augmentation budget $N$, Condition Diffusion Model, LVLM

**Ensure:** Repaired model $f_{\theta^*}$

— **Stage 1: Baseline Training & Diagnosis** —

1: Train baseline model $f_\theta$ on $\mathcal{D}_{\text{train}}$
2: Compute $\text{Acc}(\mathcal{D}_{\text{val}})$                                       ▷ Overall validation accuracy
3: **for all** attribute $a_i \in \mathcal{A}$, value $v \in \mathcal{V}_i$ **do**
4:      $S \leftarrow \{x \in \mathcal{D} \mid a_i(x) = v\}$
5:      Compute training support: $r(S) \leftarrow |S \cap \mathcal{D}_{\text{train}}| \,/\, |\mathcal{D}_{\text{train}}|$
6:      Compute slice accuracy: $\text{Acc}(S)$
7:      **if** $r(S) < \rho$ **and** $\text{Acc}(\mathcal{D}_{\text{val}}) - \text{Acc}(S) > \epsilon$ **then**
8:          Flag $S$ as bug slice $S_e$ with error attributes $\{(a_j, v_j')\}_{j \in \mathcal{J}}$
9:      **end if**
10: **end for**
11: Rank all bug slices by validation error; select top-$k$ attributes for augmentation

— **Stage 2: Targeted Generation with CDM** —

12: $\mathcal{G} \leftarrow \emptyset$                                                       ▷ Set of generated images
13: **for all** training image $x \in \mathcal{D}_{\text{train}}$ that does **not** satisfy $S_e$ **do**
14:      Construct prompt from error attributes $\{(a_j, v_j')\}_{j \in \mathcal{J}}$
15:      Extract structure map (e.g., HED soft edge) from $x$
16:      $x' \leftarrow \text{CDM}(x, \text{prompt}, \text{structure map})$                 ▷ Edit target attributes, preserve others
17:      $y(x') \leftarrow y(x)$                                          ▷ Retain original label
18:      $\mathcal{G} \leftarrow \mathcal{G} \cup \{(x', y(x'))\}$
19:      **if** $|\mathcal{G}| \geq N$ **then**
20:          **break**
21:      **end if**
22: **end for**

— **Stage 3: LVLM Filtering** —

23: $\mathcal{G}_{\text{filtered}} \leftarrow \emptyset$
24: **for all** $(x', y') \in \mathcal{G}$ **do**
25:      $pass \leftarrow$ **true**
26:      **for all** edited attribute pair $(a_j, v_j')$ in bug slice $S_e$ **do**
27:          Query LVLM: "Does this image have $a_j = v_j'$?"
28:          **if** LVLM answers **no then**
29:              $pass \leftarrow$ **false**; **break**
30:          **end if**
31:      **end for**
32:      Query LVLM: "Is this image labeled $y'$?"
33:      **if** LVLM answers **no then**
34:          $pass \leftarrow$ **false**
35:      **end if**
36:      **if** $pass$ **then**
37:          $\mathcal{G}_{\text{filtered}} \leftarrow \mathcal{G}_{\text{filtered}} \cup \{(x', y')\}$
38:      **end if**
39: **end for**

— **Stage 4: Retrain** —

40: $\mathcal{D}_{\text{train}}' \leftarrow \mathcal{D}_{\text{train}} \cup \mathcal{G}_{\text{filtered}}$                                   ▷ Augmented training set
41: Retrain model: $f_{\theta^*} \leftarrow \text{Train}(f_\theta, \mathcal{D}_{\text{train}}')$
42: Evaluate $\text{Acc}(\mathcal{D}_{\text{val}})$ and $\text{Acc}(S_e)$ on the repaired model
43: **return** $f_{\theta^*}$

---

# B   Attribute Assignment Details

**CelebA.** Attribute assignments for the CelebA dataset are based on the official annotations provided by the dataset and follow the settings used in HiBug (Chen et al., 2023).

**ImageNet10.** Since ImageNet10 does not provide attribute annotations, we defined several attribute categories. As an example, we describe three representative categories: `color`, `background`, and `texture`. Each category is associated with a predefined set of candidate values, natural language prompt templates, and corresponding visual question answering (VQA) queries directed to the BLIP model. The structure is outlined below:

> **color**
>   *Values:* red, green, pink, brown, white, blue, tan, silver, orange, gray, maroon,    yellow, multicolored, purple, black
>   *Prompt:* "A photo of a #1 #LABEL."
>   *Question:* "What color is the main object?"
>
> **background**
>   *Values:* sky, trees, inside a house, buildings, grass, rocks, bridge, water, wall,    house, street, wild, snow
>   *Prompt:* "The background of this photo is #1."
>   *Question:* "What is the background of this photo?"
>
> **texture**
>   *Values:* plastic, metallic, wooden, leather, fabric, ceramic, glass
>   *Prompt:* "A photo of a #1 #LABEL."
>   *Question:* "What texture is the main object?"

These definitions are used throughout the pipeline to assign, generate, and validate attribute-specific variations for model debugging on ImageNet10.

# C   Conditional Diffusion Model Details

We use *Stable Diffusion v1.5* as our text-to-image diffusion backbone and apply ControlNet (Zhang et al., 2023) to enable fine-grained attribute control. The ControlNet model is conditioned on soft edge maps extracted from original training images using the HED detector (Xie & Tu, 2015), which preserves structural consistency during generation.

For each rare-case slice, we construct an attribute-preserving prompt. The tokens are mapped from attribute identifiers using the following dictionary:

```
token_map = {
  'redhair': 'vibrant red hair',
  'brownskin': 'brown skin',
  'sademotion': 'sad emotion'
}
```

A final prompt is created by joining the mapped attribute phrases. For example:

> ```
> a person with vibrant red hair and brown skin, in sad emotion, not change
> other previous color, high detail, natural lighting
> ```

This prompt emphasizes the target attributes while including an explicit instruction to `not change other previous color`, helping the conditional diffusion model preserve unrelated visual aspects of the original image.

We use the following additional prompts during generation:

- Positive prompt: `best quality, extremely detailed`

- Negative prompt: `lowres, bad anatomy, bad hands`

All generated images are later filtered by a vision–language model to ensure semantic correctness before being added to the training set.

# D  Large Vision–Language Model (LVLM) Details

## D.1  LVLM Query Format

For semantic filtering of generated images, we use two large vision–language models (LVLMs), specifically Qwen2.5-VL (Bai et al., 2025) and LLaVA-v1.5 (Liu et al., 2023), to verify whether each image accurately satisfies its intended attribute-value conditions.

**CelebA.** For each generated image, we query the LVLM with a set of natural language questions tailored to the attributes of interest. For example:

- For the `lipstick` attribute:

    `Is the person in this picture wearing lipstick?`

- For other attributes (e.g., hair color or skin tone):

    `Does the person in this picture have brown skin?`

**ImageNet10.** For each image–attribute pair, we construct a set of verification questions as follows:

    Does the {class name} have {value} {attribute}?
    Is there a {class name} in the picture?

All prompts share a common instruction:

    For each question, only answer with 1 (yes) or 0 (no).  Provide answers
    separated by spaces.

These structured prompts ensure that the LVLM produces reliable binary outputs for filtering. Only images for which all responses are "1" for the selected attributes and that retain the same labels are kept for training.

## D.2  Additional Analysis for Filtering Attributes

In practice, we observed that the Qwen2.5-VL model often exhibits uncertainty when verifying emotional expressions. For some images, the emotion (e.g., `sad`) is inherently ambiguous and not reliably recognizable even by humans. In such cases, the LVLM often fails to respond decisively with a "1" or "0". This leads to a relatively low filtering pass rate for emotion-related attributes. This ambiguity reflects the intrinsic difficulty of emotion recognition, especially in the absence of strong visual cues.

Empirically, the pass rate of Qwen2.5-VL for emotion filtering is approximately 80%. In contrast, most other attributes (e.g., `color`, `skin`, `texture`) already yield filtering accuracies above 90%. When keeping the original label unchanged, the filtering results are mostly correct, with over 98% of the generated samples preserving the original label. When combining label consistency with two or three target attributes in the CelebA dataset, the overall filtering process achieves an average accuracy of approximately 70%. For the ImageNet10 dataset, where only two attributes—`color` and `texture`—are modified while keeping the label unchanged, the average filtering accuracy slightly exceeds 80%.

Table 8: Cross-model LVLM validation of generated samples.

| Dataset | Target attributes | Original filter | Verifier | Accepted | Attr. pass | Label pass | All pass |
|---------|-------------------|-----------------|----------|----------|------------|------------|----------|
| CelebA | RH + BS + SE | Qwen2.5-VL-7B | LLaVA-v1.5-7B | 1,000 | 91.8% | 98.4% | 87.6% |
| CelebA | RH + BS + SE | LLaVA-v1.5-7B | Qwen2.5-VL-7B | 1,053 | 78.6% | 97.1% | 72.4% |
| ImageNet10 | pink color + fabric texture | Qwen2.5-VL-7B | LLaVA-v1.5-7B | 1,000 | 94.2% | 99.0% | 91.5% |
| ImageNet10 | pink color + fabric texture | LLaVA-v1.5-7B | Qwen2.5-VL-7B | 1,129 | 84.7% | 98.2% | 80.3% |

Nonetheless, when using LLaVA-v1.5, the pass rate for emotion filtering exceeds 95%, while the filtering accuracy for other attributes remains around 90%. We hypothesize that emotional attributes are more difficult for conditional diffusion models to modify precisely, which in turn makes it harder for LVLMs to detect whether the intended change is present. We choose to use Qwen2.5-VL as our default LVLM throughout most experiments to ensure consistency, reproducibility, and stricter evaluation of attribute presence, especially under less ideal generation conditions.

### D.3 Cross-Model LVLM Validation

To test whether filtering decisions are artifacts of a single LVLM, we perform a cross-model validation check. Samples accepted by Qwen2.5-VL are re-evaluated by LLaVA-v1.5, and samples accepted by LLaVA-v1.5 are re-evaluated by Qwen2.5-VL. This does not require retraining and directly measures whether the retained images still satisfy the target attributes and preserve the label under an independent verifier.

Table 8 shows that LLaVA-v1.5 is less conservative and accepts more generated samples, while Qwen2.5-VL gives a stricter verification signal. Samples accepted by one LVLM still achieve high all-condition pass rates under the independent verifier, indicating that the filtering results are not only artifacts of a single LVLM's decision boundary.

## E  Additional Implementation Details

All experiments are conducted on an NVIDIA A100 GPU. We use the CelebA "wearing lipstick" classification task as a benchmark to evaluate three backbone architectures: ResNet-18 (He et al., 2016), ViT-B/16 (Dosovitskiy et al., 2020), and CLIP (ViT-B/32) (Radford et al., 2021). All models are initialized with random weights and trained from scratch. This design allows us to isolate the effectiveness of SafeFix from pre-existing biases inherent in pre-trained weights, such as the latent knowledge of ImageNet. This approach is also necessary because when generated images are added to a source dataset, the specific large-scale datasets used for subsequent model training are often unknown, and retraining on such data is not feasible. For the optimization process, we use cross-entropy loss and the Adam optimizer (Kingma & Ba, 2015) with a learning rate of $1 \times 10^{-4}$ and weight decay of $1 \times 10^{-3}$. All experiments use a batch size of 64 and are conducted for 20 epochs.

For rare-case bug discovery and augmentation, we take the CelebA "wearing lipstick" classification task as an example. We first use a computer vision model $f_\theta \colon \mathcal{X} \to \mathcal{Y}$ to produce predictions and compute the overall test accuracy $\mathrm{Acc}(\mathcal{D}_{\mathrm{val}})$. For our experiments on the CelebA dataset, we use 7 distinct hair color attributes. These color categories were initially proposed using ChatGPT (GPT-4 with vision (OpenAI, 2023)) and subsequently assigned to the images by the BLIP (Li et al., 2022) model. This assignment process followed the same setting as in (Chen et al., 2023). Accordingly, we set the rare threshold $\rho = 0.05$ (which is less than $1/7 \approx 0.143$) and the accuracy difference threshold $\epsilon = 0.03$. We found there are 2,484 red-hair images out of the 80,000 training samples, which corresponds to only 3.11% of the data. Their slice accuracy is 0.8901, which is lower than the overall $\mathrm{Acc}(\mathcal{D}_{\mathrm{val}}) = 0.9068$. Based on these thresholds, we identify the subset of red-hair images that satisfy both conditions and treat them as a candidate rare-case bug slice. For rare-case bug discovery and augmentation, we follow the procedure described above, using accuracy and slice support to identify candidate attributes. We set the rarity threshold $\rho = 0.05$ and the accuracy difference threshold $\epsilon = 0.03$ based on dataset statistics. For synthetic augmentation, we use ControlNet (Zhang et al., 2023) as the conditional diffusion model, which is based on Stable Diffusion 1.5 (Rombach et al., 2022) and conditioned

on soft HED boundaries (Xie & Tu, 2015). The soft HED boundary preserves fine structural details from the input images, making this approach particularly suitable for attribute-preserving edits such as recoloring and stylizing. To filter and validate generated images, we employ Qwen2.5-VL (Bai et al., 2025) as the primary large vision–language model (LVLM), and additionally compare results with LLaVA-v1.5 (Liu et al., 2023).

In terms of computational cost, generating 1,000 images with ControlNet requires slightly over one hour, while the subsequent filtering of these images with our LVLM takes approximately 10 minutes. From an algorithmic complexity perspective, the ControlNet generation process exhibits a computational complexity of $O(D \times S \times U)$, where $D$ represents the number of images in the dataset, $S$ denotes the number of DDIM sampling steps (in our experiments we set $S = 30$), and $U$ corresponds to the complexity of each UNet forward pass. For Stable Diffusion 1.5, each UNet forward pass requires approximately ($\approx$2.3 TFLOPs) of computation with 860M parameters operating on 64×64 latent representations. The dominant computational bottleneck lies in the iterative denoising process, where each of the 30 sampling steps requires a full forward pass through the UNet architecture.

In contrast, our LVLM-based filtering pipeline demonstrates a more efficient complexity of $O(D \times I)$, where $I$ represents the inference time per image for the vision–language model. Empirical evaluation on the ImageNet images shows that our LLaVA-v1.5-7B filtering achieves a throughput of 7,025 images per hour with an average processing time of 0.512 seconds per image. The significantly lower computational overhead of the filtering stage (approximately 7× faster) stems from the single-pass nature of LVLM inference, eliminating the iterative sampling required by diffusion models. Resource utilization analysis reveals that the filtering process maintains stable GPU memory usage at 14.7GB with 98.6% of computation time dedicated to model inference, demonstrating high computational efficiency. The filtering pipeline also achieves a 70-80% success rate in identifying images matching the specified attributes, validating both the effectiveness of our multi-attribute prompting strategy and the quality of the dataset for the target criteria. This efficiency advantage becomes increasingly pronounced when processing large-scale datasets, as the filtering complexity scales linearly with dataset size while maintaining constant per-image processing time.

**Data augmentation baseline.** We apply on-the-fly augmentation to each training image while keeping the dataset size fixed. For each image, we sample the following transformations in order:

$$\text{RRC}(224, \text{ scale} = [0.08, 1.0], \text{ ratio} = [3/4, 4/3]) \ \rightarrow \ \text{HFlip}(p = 0.5) \ \rightarrow$$
$$\text{ColorJitter}(b = 0.4, c = 0.4, s = 0.4, h = 0.1) \ \rightarrow \ \text{Grayscale}(p = 0.2) \ \rightarrow \ \text{ToTensor} \ \rightarrow$$
$$\text{RandomErasing}(p = 0.25, \text{ area} = [0.02, 0.33], \text{ ratio} = [0.3, 3.3]) \ \rightarrow$$
$$\text{Normalize}(\mu = [0.5, 0.5, 0.5], \ \sigma = [0.5, 0.5, 0.5]).$$

At test time, we use a deterministic resize and center crop followed by normalization:

$$\text{Resize}(256) \ \rightarrow \ \text{CenterCrop}(224) \ \rightarrow \ \text{ToTensor} \ \rightarrow \ \text{Normalize}(\mu, \sigma).$$

## F   Accuracy Results

In the Results section, we introduce Relative Error Reduction (RER), which can be inferred from accuracy metrics. Therefore, we include the accuracy results in Table 9 and Table 10 for the CelebA and ImageNet10 datasets, respectively.

On **CelebA**, we select rare-case bugs defined by attribute–value combinations `red hair`, `brown skin`, and `sad emotion` for ResNet and ViT, and `yellow hair`, `brown skin`, and `sad emotion` for CLIP, based on the most frequent patterns identified among failure slices. SafeFix consistently achieves the highest test accuracy across all models (ResNet, ViT, and CLIP) and varying levels of synthetic augmentation as shown in Table 9. For example, with 1,000 added images, our method improves accuracy by +1.35% (ResNet), +2.89% (ViT), and +2.78% (CLIP) relative to their respective baselines. These results show that our attribute-targeted augmentation and filtering pipeline is effective in repairing rare-case failure slices, outperforming both CDM-based and HiBug baselines.

On **ImageNet10**, similar trends emerge, as shown in Table 10. For ResNet and ViT, we target rare-case bugs involving `pink color` and `fabric texture`, while for CLIP we use `orange color` and `fabric texture`.

Table 9: Test accuracy (%) on CelebA for varying numbers of added images, models, and methods. Ours (L) and Ours (Q) denote using LLaVA-7B and Qwen-7B as the large vision–language model filter, respectively.

| Method | ResNet | | | ViT | | | CLIP | | |
|---|---|---|---|---|---|---|---|---|---|
| | 1k Images | 5k Images | 10k Images | 1k Images | 5k Images | 10k Images | 1k Images | 5k Images | 10k Images |
| Base | | 90.57 | | | 85.02 | | | 88.32 | |
| Data Augmentation | 90.80 | 90.74 | 90.87 | 85.66 | 85.40 | 85.60 | 88.45 | 88.87 | 88.59 |
| DiGA | 90.79 | 90.96 | 90.91 | 85.91 | 85.74 | 85.79 | 88.72 | 88.66 | 89.02 |
| DA-CDM | 90.95 | 90.88 | 90.86 | 85.63 | 86.55 | 86.04 | 90.11 | 90.25 | 90.14 |
| Mask-ControlNet | 91.08 | 91.26 | 91.15 | 86.71 | 86.59 | 86.95 | 90.20 | 90.18 | 90.28 |
| HiBug_Class | 91.05 | 91.12 | 90.89 | 85.79 | 86.18 | 86.10 | 90.03 | 90.07 | 89.95 |
| HiBug_Task | 91.21 | 91.03 | 90.94 | 87.82 | 86.78 | 86.18 | 90.16 | 90.09 | 90.03 |
| **Ours (L)** | 91.55 | 91.65 | 91.67 | 87.73 | **87.78** | 87.32 | 90.92 | 90.95 | **90.86** |
| **Ours (Q)** | **91.92** | **91.71** | **91.98** | **87.91** | 87.33 | **87.41** | **90.98** | **91.02** | 90.71 |

Table 10: Test accuracy (%) on ImageNet10 for varying numbers of added images, models, and methods. Ours (L) and Ours (Q) denote using LLaVA-7B and Qwen-7B as the large vision–language model filter, respectively.

| Method | ResNet | | | ViT | | | CLIP | | |
|---|---|---|---|---|---|---|---|---|---|
| | 100 Images | 500 Images | 1k Images | 100 Images | 500 Images | 1k Images | 100 Images | 500 Images | 1k Images |
| Base | | 71.73 | | | 97.42 | | | 93.78 | |
| Data Augmentation | 72.73 | 72.35 | 73.05 | 97.45 | 97.51 | 97.49 | 94.04 | 93.88 | 94.02 |
| DiGA | 72.34 | 73.33 | 72.86 | 97.53 | 97.50 | 97.58 | 94.12 | 94.05 | 93.95 |
| DA-CDM | 73.21 | 73.80 | 74.14 | 97.58 | 97.76 | 97.77 | 93.96 | 94.05 | 94.15 |
| Mask-ControlNet | 72.22 | 72.54 | 72.39 | 97.25 | 97.69 | 97.52 | 93.44 | 93.88 | 93.70 |
| HiBug_Class | 72.24 | 72.15 | 71.94 | 97.12 | 97.07 | 97.28 | 93.92 | 94.21 | 93.83 |
| HiBug_Task | 73.51 | 72.88 | 73.36 | 97.66 | 97.78 | 97.58 | 93.29 | 94.18 | 94.15 |
| **Ours (L)** | 74.10 | **74.19** | 73.92 | 98.21 | **98.17** | 98.11 | 94.49 | 94.80 | 94.44 |
| **Ours (Q)** | **74.31** | 73.43 | **74.88** | **98.22** | 98.09 | **98.41** | **94.57** | **94.88** | **94.95** |

Across all models, our proposed method consistently surpasses the baseline methods. Specifically, our method improves ResNet accuracy by +3.18% (at 100 images) compared to the base model. ViT and CLIP also exhibit a steady improvement compared to other methods.

# G  Large-Scale Settings in Main Results

In the main experiments, we show that SafeFix improves test accuracy by augmenting underrepresented slices. To further assess its effectiveness under large-scale augmentation, we consider a scenario where a previously rare-case attribute—`red hair`—becomes common in training after adding 10,000 generated images (around 12,480 red-hair images out of 90,000 total), while its prevalence in the fixed validation set remains low (621 out of 20,000).

This setup creates an important distinction: although `red hair` is no longer rare in training ($\sim 14\%$), it remains rare in validation ($\sim 3\%$). Nevertheless, the accuracy on the red-hair subset improves significantly, demonstrating that SafeFix effectively improves generalization to previously underperforming groups.

**Interpretation.**  This setup aligns with the goal of rare-case debugging: the term "rare case" refers to slices that are rare and inaccurate in evaluation or real-world deployment. The objective is to increase their representation in training to address model failure. Even though `red hair` is no longer rare in training, the model successfully reduces its failure rate in evaluation—without changing the validation distribution.

Our fixed validation set continues to reflect real-world statistics, ensuring that improvements in rare-case performance are meaningful. This practice—augmenting rare cases in training while keeping the test distribution fixed—is consistent with standard methodology in prior work (Chen et al., 2023). These findings

Table 11: Comparison with targeted generation repair baselines.

| Method | CelebA Acc. | CelebA Fixed Bugs | ImageNet10 Acc. | ImageNet10 Fixed Bugs |
|---|---|---|---|---|
| Base | 0.9057 | – | 0.7173 | – |
| Dataset Interfaces (DI) | 0.9132 | 8/13 | 0.7297 | 13/17 |
| Latent-space Failure Directions (LFD) | 0.9188 | **11/13** | 0.7454 | **15/17** |
| **SafeFix** | **0.9192** | 10/13 | **0.7488** | **15/17** |

Table 12: Comparison with long-tail learning baselines using only original training data.

| Method | CelebA Acc. | CelebA Fixed Bugs | ImageNet10 Acc. | ImageNet10 Fixed Bugs |
|---|---|---|---|---|
| Base | 0.9057 | – | 0.7173 | – |
| Focal Loss | 0.9134 | 8/13 | 0.6880 | 11/17 |
| Class-Balanced Loss | 0.9122 | 8/13 | 0.7270 | 13/17 |
| Oversampling | 0.9101 | 7/13 | 0.7290 | 14/17 |
| LDAM | 0.9141 | 8/13 | 0.7120 | 12/17 |
| **SafeFix** | **0.9192** | **10/13** | **0.7488** | **15/17** |

confirm that SafeFix improves generalization on rare-case bugs without sacrificing reliability or introducing new biases.

# H  Additional Experiments

## H.1  Comparison with Targeted Generation Repair Baselines

We compare SafeFix with two targeted repair baselines that also use generated or failure-directed data. Dataset Interfaces (DI) (Vendrow et al., 2023) is adapted from counterfactual diagnosis to repair by using its generated counterfactual samples as augmentation data and retraining the downstream classifier under the same data budget as SafeFix. For ImageNet10, DI learns one class-specific textual inversion token per class and generates attribute-conditioned counterfactual samples with prompts such as "a photo of `<class-token>` with pink color and fabric texture". We also adapt Latent-space Failure Directions (LFD) (Jain et al., 2023a): for each class, LFD trains a linear SVM in CLIP image-embedding space to separate correctly classified validation samples from misclassified ones, then uses the learned normal vector as a class-specific failure direction for generation.

Table 11 shows that the adapted baselines are strong, especially LFD. LFD fixes slightly more CelebA bugs because it explicitly learns directions that separate misclassified from correctly classified validation samples, so its generations concentrate near the model's failure modes. SafeFix still achieves the highest test accuracy on both datasets and matches the best ImageNet10 bug-fixing count, indicating a better repair–accuracy tradeoff.

## H.2  Comparison with Long-Tail Learning Baselines

Rare-case repair is related to long-tail learning, but the failure mode is different. Standard long-tail learning methods operate on class imbalance, whereas our bug slices are attribute-level subgroups such as `red hair` + `brown skin` in a lipstick classifier. We therefore evaluate four long-tail baselines using only the original training data: Focal Loss (Lin et al., 2017), class-balanced loss (Cui et al., 2019), oversampling, and LDAM (Cao et al., 2019).

As shown in Table 12, class-level reweighting and resampling can improve some aggregate metrics, but they do not reliably repair attribute-level subpopulation failures. SafeFix improves overall accuracy while fixing more discovered attribute-level bugs, because it augments the diagnosed semantic slice directly rather than only rebalancing class labels.

### H.3 Generation Quality Analysis

We compare HiBug and SafeFix using four key metrics—LPIPS diversity (Zhang et al., 2018), CLIP consistency (Radford et al., 2021), Fréchet Inception Distance (FID) (Heusel et al., 2017) against real images, and the KL divergence of deep-feature distributions—using the ImageNet10 dataset with the `pink color +` `fabric texture` setting as an illustrative example, as shown in Table 13.

Table 13: Generation-quality metrics for HiBug vs. SafeFix on ImageNet10 (pink-color + fabric-texture). "Real->Gen" indicates the Frechet distance computed between the real and generated image feature distributions; lower FID means the generated set is closer to the real distribution. Arrows indicate preference: ↑ higher is better; ↓ lower is better.

| Metric | HiBug_Task | SafeFix |
|---|---|---|
| LPIPS diversity ↓ | 0.7582 | **0.6764** |
| CLIP consistency ↑ | **0.2087** | 0.2049 |
| FID (Real->Gen) ↓ | 0.2405 | **0.1490** |
| KL divergence ↓ ($\times 10^6$) | 7.05 | **5.71** |

SafeFix achieves substantially lower FID and KL divergence compared to HiBug, indicating that its samples lie much closer to the real-image distribution. Although its LPIPS diversity is moderately reduced and CLIP consistency slightly lower, these modest trade-offs yield more reliable, attribute-faithful examples for downstream repair. In practice, fewer but higher-quality variants help correct rare-case failures without introducing noisy or out-of-distribution artifacts.

### H.4 SafeFix with alternative conditional diffusion models

The targeted generation stage in our pipeline uses ControlNet (Zhang et al., 2023) as the conditional diffusion model. Although ControlNet is effective due to its stable structure-preserving behavior, relying on a single generator raises questions about whether SafeFix depends on ControlNet-specific features. To test the generality of the pipeline, we replaced ControlNet with another conditional diffusion model, InstructPix2Pix (Brooks et al., 2023), while keeping the LVLM filter (Qwen2.5-VL). The resulting accuracy on CelebA, shown in Table 14, exceeds all baselines reported in the main results and is only slightly lower than SafeFix with ControlNet. This demonstrates that the pipeline is not tied to ControlNet and functions well with alternative generators.

### H.5 Qualitative Analysis of Attribute Editing and Failure Cases

Figure 3 shows some good examples generated by the conditional diffusion model, but conditional diffusion models can make mistakes. In the CelebA dataset, we primarily target rare-case attributes such as red hair (RH), brown skin (BS), and sad emotion (SE), but our framework is flexible and supports other attribute configurations. Figure 6 shows examples of original and SafeFix-generated images. In the first example, only yellow hair was added to the original face while all other identity features remained unchanged. The second image reflects a transformation into the red hair and brown skin attribute combination. In the third example, the model aimed to edit the hair color to red, but instead altered the background to a reddish hue without changing the hair itself—highlighting a typical failure of generative models when not filtered.

Figure 7 shows attribute editing results on ImageNet10 for the pink color and fabric texture combination. In the first example, a multicolored leather purse is successfully transformed into a pink fabric purse. The second example shows a black plastic backpack converted into a pink fabric version, though the modified velcro strap appears slightly unnatural—suggesting semantic correctness but imperfect realism. In the third case, a black leather barber chair becomes pink, but its texture remains visibly leather-like, indicating that not all attributes were successfully edited.

While over 90% of generated images (as verified by Qwen2.5-VL) accurately reflect the target attributes, occasional failures illustrate that conditional diffusion models (CDMs) can still make subtle mistakes. These

Table 14: RER (%) using different conditional diffusion models in the targeted generation stage. Instruct-Pix2Pix (Brooks et al., 2023) serves as an alternative generator and shows strong repair performance.

| Method | ResNet (base acc: 90.57%) | | |
|---|---|---|---|
| | 1k Images | 5k Images | 10k Images |
| DA-CDM | 4.03 | 3.29 | 3.08 |
| Mask-ControlNet | 5.41 | 7.32 | 6.15 |
| HiBug_Class | 5.09 | 5.83 | 3.40 |
| HiBug_Task | 6.79 | 4.88 | 3.92 |
| **Ours (InstructPix2Pix)** | 11.32 | 11.85 | 11.74 |
| **Ours (ControlNet)** | **14.32** | **12.09** | **14.95** |

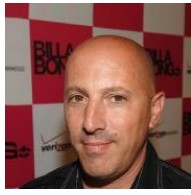 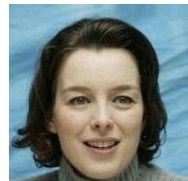 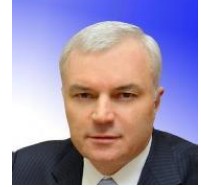 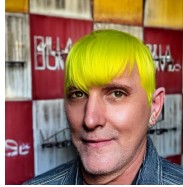 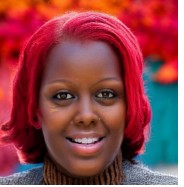 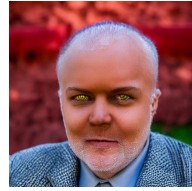

(a) Original CelebA samples         (b) Attribute-modified images from SafeFix

Figure 6: Comparison between original CelebA images (left) and attribute-edited outputs (right) generated by SafeFix.

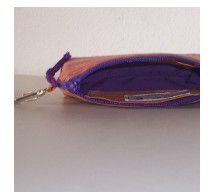 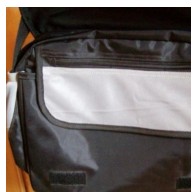 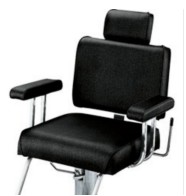 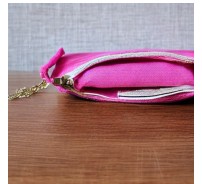 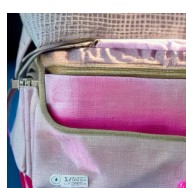 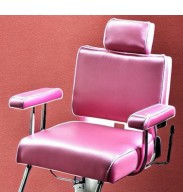

(a) Original ImageNet10 samples        (b) Attribute-modified images from SafeFix

Figure 7: Comparison between original ImageNet10 images (left) and attribute-edited outputs (right) by SafeFix targeting pink color and fabric texture.

examples are representative failure cases selected for analysis; most generated images do not exhibit such issues. The majority of CelebA and ImageNet10 samples are correctly modified, but these rare errors underscore the importance of LVLM-based filtering to ensure semantic correctness and support reliable model repair.

## H.6 Evaluating Synthetic Data as a Standalone Training Set

In our main experiments, we added $N$ validated synthetic images to $\mathcal{D}_{\text{train}}$ while keeping $\mathcal{D}_{\text{val}}$ and $\mathcal{D}_{\text{test}}$ fixed. To further examine the end-to-end value of our augmentation, we instead randomly split the $N$ synthetic images into an 8:1:1 train–val–test partition and trained ResNet from scratch. As shown in Table 15, our method consistently outperforms HiBug by large margins across all settings. This suggests that SafeFix, using a conditional diffusion model, generates images with more internally coherent distributions—that is, the synthetic samples are more semantically consistent with each other. It also shows that LVLM filtering reliably ensures label correctness, resulting in improved downstream classification accuracy. Together, these two key components—the CDM and the LVLM—underscore the semantic fidelity and effectiveness of our augmentation strategy.

Table 15: Test accuracy (%) on $N$ synthetic images, trained from scratch. The $N$ images were split into an 8:1:1 train–val–test partition for this experiment.

| # Images | HiBug_Task | **Ours** |
|---|---|---|
| 1,000 | 63.81 | **81.95** |
| 5,000 | 76.44 | **90.54** |
| 10,000 | 83.03 | **90.87** |

Table 16: Effect of rare threshold $\rho$ on the number of identified bugs before and after repair.

| **Dataset** | $\rho$ | **Pre** | **Post** |
|---|---|---|---|
| CelebA | 0.01 | 3 | 2 |
| CelebA | 0.05 | 10 | 4 |
| CelebA | 0.06 | 12 | 4 |
| CelebA | 0.07 | 12 | 4 |
| CelebA | 0.08 | 12 | 4 |
| CelebA | 0.14 | 15 | 9 |
| ImageNet10 | 0.01 | 4 | 3 |
| ImageNet10 | 0.05 | 13 | 7 |
| ImageNet10 | 0.066 | 16 | 9 |

Table 17: Effect of accuracy-difference threshold $\epsilon$ on the number of identified bugs before and after repair.

| **Dataset** | $\epsilon$ | **Pre** | **Post** |
|---|---|---|---|
| CelebA | 0.01 | 6 | 3 |
| CelebA | 0.02 | 8 | 3 |
| CelebA | 0.03 | 10 | 4 |
| CelebA | 0.10 | 13 | 9 |
| ImageNet10 | 0.01 | 7 | 4 |
| ImageNet10 | 0.02 | 9 | 5 |
| ImageNet10 | 0.03 | 13 | 7 |
| ImageNet10 | 0.10 | 15 | 12 |

### H.7 Threshold settings for identifying rare-case slices

Settings for the rare threshold ($\rho$) and the accuracy-difference threshold ($\epsilon$) are dataset-dependent. As shown in Table 16, with $\rho = 0.05$, the number of identified bugs drops from 10 to 4 on CelebA and from 13 to 7 on ImageNet10. For CelebA, increasing $\rho$ to 0.06, 0.07, and 0.08 yields the same result, which raises the pre-fix bug count to 12, while the post-fix count remains stable at 4. Extreme values of $\rho$ become invalid: for ImageNet10, which contains 15 color categories, a value such as $\rho = 0.2$ exceeds the 1/15 ratio of each color and thus no longer reflects a rare case.

The accuracy-difference threshold $\epsilon$ follows a similar trend. Small values isolate a limited set of failure slices, while moderate values increase the number of identified bugs without affecting the repaired result. Table 17 shows that using $\epsilon \in \{0.01, 0.02, 0.03\}$ yields stable post-fix counts on both datasets. Extreme thresholds such as $\epsilon = 0.1$ become unreliable because they classify many non-critical cases as failures, inflating both pre- and post-fix bug counts.

## I  Future Work

SafeFix demonstrates that *targeted* synthetic augmentation plus LVLM filtering can reliably repair rare–case bugs across image classification and denser visual prediction tasks. Several promising research directions remain:

- **Data–selection vs. data–generation.** HiBug2 (Chen et al., 2025) focuses on *discovering* failure slices and then *retrieving* real images from the web to rebalance those slices instead of *generating* images. A natural extension is to combine the web–retrieval paradigm with SafeFix's LVLM verifier—e.g. apply the same binary / attribute QA filter to crawled images before retraining. This may further reduce covariate shift without the risk of generative artifacts.

- **Broader modalities and tasks.** Our experiments already cover classification, pose estimation, and object detection. Extending SafeFix to image captioning, VQA, video, and text–image inputs is an exciting avenue, especially for reasoning-centric benchmarks.

- **Beyond discrete attributes.** Discrete vocabularies of attributes are common failure sources, hence our focus. Future work can extend this direction by handling continuous or fine-grained attributes, supporting multi-attribute interactions, and studying how these repair signals transfer across datasets and model families.

**Societal Impact.** The proposed method could be important for real-world applications such as fairness auditing and reliability improvement in vision systems deployed in robotics, healthcare, surveillance, and auto-driving, by automatically identifying and repairing failure modes associated with underrepresented or biased visual patterns.

Overall, SafeFix provides a foundation for a general *data-centric* loop: **diagnose** $\rightarrow$ **generate / retrieve** $\rightarrow$ **filter** $\rightarrow$ **retrain**. Enhancing each stage with active selection, dynamic attributes, and multi-task diffusion backbones represents a rich future research agenda.

