# OpenReview forum: "SafeFix: Targeted Model Repair via Controlled Image Generation"
_TMLR — Accepted by TMLR_

### Review · Reviewer_1vpz · 2026-03-18

**Summary Of Contributions:**

SafeFix is a pipeline for fixing vision classifiers that fail on rare attribute subgroups. It has four stages:
1. diagnose rare-case failure slices using HiBug-style attribute-based analysis,
2. generate synthetic images for those slices using ControlNet conditioned on HED edges from real images,
3. filter generated images with an LVLM (Qwen2.5-VL) to verify attribute correctness,
4. augment the training set and retrain. Experiments on CelebA (binary lipstick classification) and a 10-class ImageNet subset show Fix Rate improvements over six baselines across ResNet, ViT, and CLIP.

Once the failing subgroup is identified (via HiBug), the remaining task is generating more data for a rare slice and retraining, which is similar to long-tail learning with synthetic augmentation. The paper's contribution over HiBug is using a better generator (ControlNet instead of unconditioned text-to-image) and adding a quality filter (LVLM).

Strengths:
- The LVLM filtering step addresses a real problem: diffusion models frequently get target attributes wrong.
- Section 5.4 (red-hair vs. yellow-hair augmentation on CLIP) shows targeting the correctly diagnosed bug matters.
- Clear writing, clean algorithm box, good visual comparison.

Weaknesses:
- The contribution reduces to two incremental upgrades over HiBug: swap the generator, add a filter.
- The paper does not engage with the long-tail learning field. And the scalability is not well presented.

**Audience:**

Yes

**Audience Explanation:**

The problem, fixing classifiers on underrepresented subgroups, is important for fairness and deployment reliability. Using LVLMs as quality gates for synthetic augmentation is a timely idea. The diagnose-generate-filter-retrain pipeline is feasible and could be adopted by practitioners. Researchers in subgroup robustness, data-centric AI, long-tail learning, and fairness-aware training would find the framing relevant.

**Broader Impact Concerns:**

The pipeline generates synthetic faces with targeted demographic attributes (skin tone, hair color, gender). But if the LVLM rejects brown-skin images at a higher rate than white-skin images (plausible given known biases in vision-language models), the "fairness repair" pipeline may itself be biased. Similarly for the intrinsic bias in the generation model.

**Claims And Evidence:**

Yes

**Claims Explanation:**

1. The contribution is not clearly separated from prior work
The diagnosis stage (Stage 1) is HiBug's algorithm. The retraining stage (Stage 4) is standard. The actual difference is in Stages 2–3: a stronger generation backbone (ControlNet instead of plain text-to-image) and the addition of LVLM filtering. The delta over HiBug is primarily "use stronger off-the-shelf tools", which is an incremental upgrade enabled by today's foundation models.

2. The paper does not engage with the long-tail learning field. And the scalability is not well presented.
Once HiBug identifies a rare failing slice, the remaining challenge reduces to: how best to augment an underrepresented subgroup? This is fundamentally a long-tail learning problem, yet the paper makes no connection to this well-established field. In particular, the paper does not adequately analyze whether injecting synthetic data for rare slices shifts the effective training distribution and degrades performance on common classes. By the standards of the long-tail learning literature (which is concerned precisely with improving subgroup performance without compromising majority-class accuracy, and is thus essentially distinct from few-shot learning), the scale of evaluation presented here is insufficient to convincingly demonstrate distributional safety. In particular, both datasets are not large enough to support the paper's generality and scalability: CelebA is reduced to a single binary task with a small attribute vocabulary, and ImageNet10 contains only 10 hand-picked classes. This is particularly concerning given the long-tail framing, where the challenge scales with the number of classes and the complexity of attribute interactions. Demonstrating the pipeline on a setting with a larger tail and richer attribute space is necessary to make the claims convincing.

**Requested Changes:**

1. Clarify the paper's contribution.
2. Engage with the long-tail learning literature (or discussing why not), since the core task after diagnosis is augmenting underrepresented subgroups without degrading majority-class performance, which is the central problem of that field.

---

> ### Author Response · Authors · 2026-06-09
> **Rebuttal to Reviewer 1vpz**
>
> We thank the reviewer for the thoughtful review and constructive suggestions. We respond to the main points below.
>
> ## 1. Contribution Clarity
>
> **Reviewer 1vpz Comment:**
> > The contribution over prior work, especially HiBug, is not sufficiently clear.
>
> **Response:**
> We agree that Stage 1 builds on HiBug-style diagnosis and that Stage 4 is standard retraining. The contribution is the **repair mechanism in Stages 2-3**: SafeFix combines structure-conditioned editing and LVLM verification to address a failure mode of prior generation-based repair pipelines.
>
> HiBug's text-prompt generation can change non-target attributes, backgrounds, and image structure, which creates semantic drift and distribution mismatch. SafeFix anchors generation on real training images through ControlNet, edits the diagnosed failure attributes, and preserves source-image structure and unrelated visual factors. The LVLM filter then rejects generations that miss the intended attributes or change the label. We will revise the introduction and related-work discussion to state this contribution more explicitly and add a comparison table separating SafeFix from HiBug, retrieval-based repair methods, and generic augmentation methods.
>
> ## 2. Connection to Long-Tail Learning (LTL) and Distributional Safety
>
> **Reviewer 1vpz Comment:**
> > The paper should discuss long-tail learning and possible distribution shifts from synthetic repair data.
>
> **Response:**
> We agree that the long-tail learning connection should be discussed. The key distinction is between *class-level* long-tail learning and *attribute-level* subgroup failures. Standard LTL methods reweight or resample based on class frequency, but they do not see latent attribute failures such as "red hair + brown skin" in a "lipstick" classifier; they only see the main class label.
>
> To test this, we evaluated four standard LTL baselines using only the original data:
>
> | Method | CelebA Test ACC | CelebA Fixed Bugs (out of 13) | ImageNet Test ACC | ImageNet Fixed Bugs (out of 17) |
> | :--- | :--- | :--- | :--- | :--- |
> | **Base** | 0.9057 | - | 0.7173 | - |
> | Focal Loss | 0.9134 | 8 | 0.6880 | 11 |
> | Class-Balanced Loss | 0.9122 | 8 | 0.7270 | 13 |
> | Oversampling | 0.9101 | 7 | 0.7290 | 14 |
> | LDAM | 0.9141 | 8 | 0.7120 | 12 |
> | **SafeFix (Ours)** | **0.9192** | **10** | **0.7488** | **15** |
>
> SafeFix improves overall accuracy while fixing more discovered attribute-level bugs. We will **add this LTL comparison and discussion** to the revised manuscript.
>
> ## 3. Scalability and Dataset Complexity
>
> **Reviewer 1vpz Comment:**
> > The evaluation scale may be too limited to support the paper's generality claim.
>
> **Response:**
> We understand the concern about evaluation scale. We will move the appendix visual-task results into the main text, covering pose estimation and detection with larger splits and denser outputs.
>
> | Task | Dataset / Model | Train/Val/Test | Metric | Base | Best | SafeFix | Gain |
> | :--- | :--- | :--- | :--- | :--- | :--- | :--- | :--- |
> | Pose | COCO keypoints / Keypoint R-CNN R50-FPN | 20,000/1,173/1,173 | AP | 0.5902 | 0.6145 | **0.6280** | +0.0135 |
> | Detection | KITTI Car+Pedestrian / YOLOv8n | 5,000/1,240/1,241 | mAP50-95 | 0.5626 | 0.5713 | **0.5790** | +0.0077 |
>
> SafeFix outperforms both the base model and the best baseline on both tasks. We will revise the claim to present SafeFix as a **targeted model-repair framework across classification, pose estimation, and object detection**.
>
> ## 4. Broader Impact Concerns
>
> **Reviewer 1vpz Comment:**
> > LVLM filtering may introduce bias if rejection rates differ across subgroups.
>
> **Response:**
> We agree that LVLM bias is a real risk. **Appendix D.2** reports that the pass rate of Qwen2.5-VL for emotion filtering is about 80%. We therefore use a conservative filtering prompt. For emotion filtering, for example, the prompt asks the LVLM: *"If you see any hint of [target] emotion in this picture, answer 1; only answer 0 if you are completely sure there is no such emotion."* This reduces unwarranted rejections. In our experiments, using this conservative prompt instead of a stricter one does not materially change the final repaired-model accuracy.
>
> If rejection rates are higher for a subgroup, the pipeline can generate a larger initial pool to reach the target number of accepted samples; this increases compute, but it does not require accepting lower-quality images. We will **add this limitation to the Broader Impact section** and discuss debiased LVLMs and adaptive thresholds as future work.

---

> > ### Comment · Reviewer_1vpz · 2026-06-22
> >
> > Thank you for the thorough rebuttal and the new experiments. The response effectively addresses my primary concerns. 1. The distinction between class-level LTL and attribute-level failures is well-reasoned. Furthermore, framing the pipeline as a fix for semantic drift establishes the delta over HiBug., 2. Showcasing COCO and KITTI demonstrates task versatility.
> >
> > Given the empirical additions and clarified theoretical positioning, I have updated my score.

---

> > > ### Comment · Action_Editor_1UUt · 2026-06-25
> > >
> > > Hi 1vpz ,
> > >
> > > Thanks for your engagement. Can you also please enter in an official recommendation? You should be able to do this at the top of the page. The system won't let things proceed without the recommendations entered (which is separate from the initial reviews).
> > >
> > > Thanks,
> > >
> > > AE

---

### Review · Reviewer_KtnS · 2026-04-30

**Summary Of Contributions:**

The paper proposes a pipeline to improve classifier performance on rare semantic attribute compositions by generating additional samples with the rare compositions and training the model on these samples.

**Audience:**

Yes

**Audience Explanation:**

Yes, this work is definitely of interest to ML audience.

**Claims And Evidence:**

Yes

**Claims Explanation:**

The claim is supported, but it is not a contribution. I am also not sure about its novelty.

To my understanding, the claim in the paper is the design of the pipeline to improve the performance of classifiers on rare semantic subpopulations. But the pipeline is almost directly from prior works, with some modifications to strengthen individual components. For example, compared to HiBug (Chen et al., 2023), this work converts the generation process to a more controllable one w.r.t. the attributes using another prior work, ControlNet. While the use of ControlNet is still fine as a reusable module that is not claimed as part of the contributions, the overall pipeline feels like a mix of existing works. Another example of a reused idea is filtering using LVLMs. This is also a well-known idea, particularly in dataset generation.

**Requested Changes:**

1. I am still okay with this work if the pipeline can be sufficiently justified as a contribution. If I have misunderstood the prior works and their connection to this manuscript, please clarify it in the paper, preferably using a table with columns showing ways in which the manuscript differs from the prior works. In the table, please mention the high-level ways in which the proposed pipeline differs from those in the prior works (instead of "they use text-to-image models, while we use image-conditioned models").

2. According to the paper, the metric Fix Rate "measures the fraction of previously misclassified samples that are corrected." However, Eq. (6) does not compute that. It looks only at the aggregate accuracy, not per-sample accuracy.

3. This is a question, not a requested change. You mentioned that works like AdaVision and DCD retrieve samples from another dataset. If you have a sufficiently rich dataset, how do such methods compare to the proposed generation-based approach?

---

> ### Author Response · Authors · 2026-06-09
> **Rebuttal to Reviewer KtnS**
>
> We thank the reviewer for the careful reading and constructive feedback. We respond to the main concerns below.
>
> ## 1. Contribution and Novelty
>
> **Reviewer KtnS Comment:**
> > The claim is supported, but it is not a contribution. I am also not sure about its novelty. [...] the overall pipeline feels like a mix of existing works.
>
> **Response:**
> SafeFix does use existing modules such as ControlNet and large vision-language models (LVLMs). The contribution is not that these modules are new in isolation; it is the repair procedure that combines them to target underrepresented semantic subpopulations while controlling distribution shift and semantic drift.
>
> Previous generation-based methods (such as HiBug) rely solely on language prompts, which often fail to preserve attributes unrelated to the intended edit and introduce a mismatch with the original data distribution. Previous retrieval-based methods (such as AdaVision and DCD) rely on external datasets, which often introduce a domain gap.
>
> SafeFix addresses these limitations in two concrete ways:
> 1. It anchors generation on real training instances using structure-conditioned control, so the edited samples stay close to the source distribution.
> 2. It uses LVLM filtering to check that the generated images contain the intended attributes and preserve the label.
>
> We will add the following table to make this distinction explicit:
>
> | Feature | AdaVision / DCD | HiBug | SafeFix (Ours) |
> | :--- | :--- | :--- | :--- |
> | **Strategy** | Retrieval from external datasets | Generation via language prompts | Generation conditioned on training images |
> | **Distribution Alignment** | Low (Domain gap from external data) | Low (Generative artifacts and drift) | High (Preserves original image structure) |
> | **Attribute Preservation** | N/A | Low (Prompt-only control) | High (Structure conditioning) |
> | **Output Verification** | Implicit | None | Explicit (LVLM filtering) |
>
> ## 2. Metric Definition (Fix Rate)
>
> **Reviewer KtnS Comment:**
> > According to the paper, the metric Fix Rate "measures the fraction of previously misclassified samples that are corrected." However, Eq. (6) does not compute that. It looks only at the aggregate accuracy, not per-sample accuracy.
>
> **Response:**
> We appreciate this precise observation. If we expand the accuracy terms as $Acc_{\text{before}} = \frac{C_{\text{before}}}{N}$ and $Acc_{\text{after}} = \frac{C_{\text{after}}}{N}$, where $C$ is the number of correct predictions and $N$ is the total number of samples, then:
> $$ FR = \frac{Acc_{\text{after}} - Acc_{\text{before}}}{1 - Acc_{\text{before}}} = \frac{\frac{C_{\text{after}}}{N} - \frac{C_{\text{before}}}{N}}{1 - \frac{C_{\text{before}}}{N}} = \frac{\frac{C_{\text{after}} - C_{\text{before}}}{N}}{\frac{N - C_{\text{before}}}{N}} = \frac{C_{\text{after}} - C_{\text{before}}}{N - C_{\text{before}}} $$
> Here, $N - C_{\text{before}}$ is the number of previously misclassified samples, and $C_{\text{after}} - C_{\text{before}}$ is the net increase in correct predictions. In this aggregate sense, $FR$ measures the fraction of the original errors removed by repair.
>
> We agree, though, that the current wording can be clearer. Since the formula uses aggregate counts rather than per-sample transitions, we will describe it as the "relative reduction in the error rate" in the revision.
>
> ## 3. Generation-based vs. Retrieval-based Approaches
>
> **Reviewer KtnS Comment:**
> > This is a question, not a requested change. You mentioned that works like AdaVision and DCD retrieve samples from another dataset. If you have a sufficiently rich dataset, how do such methods compare to the proposed generation-based approach?
>
> **Response:**
> This is a useful question. If an infinitely rich external dataset existed, retrieval would be attractive because it avoids generative artifacts. In practice, controlled generation has several advantages:
>
> 1. **Targeting Rare Combinations:** Even in extremely large datasets, specific combinations of attributes (for example, a specific object in a highly unusual background or lighting condition) may simply not exist or remain statistically rare. Generation allows us to create these exact rare compositions on demand.
> 2. **Distribution consistency:** Retrieved samples from an external dataset can carry domain shifts such as different sensors, resolutions, or object framings. SafeFix conditions on the original training images, so non-targeted attributes and image structure remain tied to the target dataset.
> 3. **Search efficiency:** Finding rare attribute combinations in a large external pool can require scanning millions of images. Conditional generation creates the requested combination directly.

---

> > ### Comment · Reviewer_KtnS · 2026-06-14
> > **Reviewer response**
> >
> > Question 3 is adequately answered.
> >
> > I still have reservations about the contribution (Question 1). As reviewer 1vpz also rightly pointed out, the contributions over HiBug are minor. As I mentioned in my original review, the overall pipeline is the same, although the use of conditional generative models and LVLM filtering has definitely improved the accuracy (Tab. 6). That is, "the repair procedure" is not new. Also, shouldn't LVLM filtering fix the issue of the bad generated samples in HiBug (e.g., the images shown in Fig. 3(b))? Why is Hi-Bug+LVLM filtering performance low in Tab. 6? Is it because Hi-Bug and SafeFix use different generative models under the hood?
> >
> > The definition of Fix Rate must be clarified. The term "Fix Rate" itself is a bit of a misnomer. Also, the authors can edit the PDF during the discussion period.

---

> > > ### Author Response · Authors · 2026-06-20
> > > **Response to Reviewer KtnS Follow-up**
> > >
> > > We thank the reviewer for confirming that Question 3 is adequately answered. We address the remaining questions below:
> > >
> > > ### 1. Contribution Relative to HiBug (Question 1)
> > > We agree that SafeFix follows the same high-level repair structure as HiBug: diagnose a failure slice, generate additional data, and retrain. However, our contribution is in the **repair-data construction step**. Specifically, SafeFix addresses two failure modes of prompt-only generation: **semantic drift** and **distribution mismatch**.
> > > * **HiBug** proposes attribute text and generates a completely new image from the prompt. Even when the target attribute appears, the generated image may no longer match the original training distribution (e.g., changes in eye size, face contour, background, pose, and lighting).
> > > * **SafeFix** instead **conditions generation on real training images**, editing diagnosed attributes while preserving all other visual factors of the source image.
> > >
> > > ### 2. Performance of HiBug + LVLM Filtering
> > > The reviewer asks why LVLM filtering does not resolve the issue of bad generated samples in HiBug (e.g., Fig. 3(b)), and why HiBug+LVLM performance remains low in Table 6.
> > > * **Same Backbone:** Both SafeFix and HiBug use the same underlying diffusion backbone (**Stable Diffusion 1.5**). The performance difference is not due to the generative models.
> > > * **Filtering vs. Conditional Generation:** LVLM filtering can remove visibly invalid HiBug samples, but it **cannot enforce consistency with a specific original training image**. For example, even if the filter keeps an image with red hair, that image is not an edit of the source image. Non-target features (e.g., face shape, background) may all differ from the original, which are exactly the factors that prompt-only HiBug generation ignores.
> > > * **Why SafeFix Succeeds:** SafeFix performs conditional image editing from real training images and then filters the edited outputs. Thus, the LVLM operates on candidates that are already tied to the source distribution. Filtering alone cannot replace conditional generation from the original training data.
> > >
> > > ### 3. Metric Renaming and Clarification
> > > We agree that the term "Fix Rate" can be misleading. We have renamed it to **Relative Error Reduction (RER)**. The metric is computed from aggregate accuracy before and after repair:
> > >
> > > $$
> > > RER =
> > > \frac{Err_{\text{before}} - Err_{\text{after}}}
> > > {Err_{\text{before}}}
> > > = \frac{(1 - Acc_{\text{before}}) - (1 - Acc_{\text{after}})}
> > > {1 - Acc_{\text{before}}}
> > > = \frac{Acc_{\text{after}} - Acc_{\text{before}}}
> > > {1 - Acc_{\text{before}}}.
> > > $$
> > > Here, $Err = 1 - Acc$. This measures the fraction of the original error rate removed by repair. We will also report it together with absolute accuracy so the effect size is clear.
> > > Since the formula uses aggregate accuracy values rather than per-sample correctness changes, we will avoid saying that it directly counts previously misclassified samples that become correct.

---

> > > > ### Comment · Reviewer_KtnS · 2026-06-20
> > > > **Reviewer's response**
> > > >
> > > > I once again thank the authors for their work.
> > > >
> > > > Their latest response has addressed my concerns. Please be sure to include the above-mentioned differences between SafeFix and HiBug in the paper.

---

> > > > > ### Author Response · Authors · 2026-06-20
> > > > >
> > > > > We thank the reviewer for the feedback. We will include the discussion of the differences between SafeFix and HiBug in the revised manuscript.

---

### Review · Reviewer_NMkE · 2026-05-30

**Summary Of Contributions:**

This paper introduces SafeFix, which introduces a pipeline for classifier models against failures related to underrepresented data, which is mentioned as "rare-cases" in the existing dataset. The method consists of multiple steps. Initially, the method identifies classifier failures that can be characterized as high-error and rare cases w.r.t. training data, using attributes proposed by a VLM and VQA-based assessment. Folliwing this step, for the cases that satisfy the failure mode of high-error and low freqnecy, the authors propose a targeted augmentation step using ControlNet with soft HED boundaries as conditioning, which aims to preserve original distribution of the training data, rather than inserting arbitrary samples from the target attribute. Third, the generated images are filtered with LVLMs as a verification step for edit accuracy and preservation of labels. The overall claim of the paper states that since the proposed augmentation prevents degregadion of the overall accuracy of the classifier by preserving the original data distribution except the target attribute (the rar attibute targeted by the augmentation) and does not introduce any new bugs, where systematic classifier errors are defined as bugs by the authors. Authors validate the effectiveness of SafeFix on CelebA (lipstick classification), and 10-class ImageNet subset across ResNet-18, Vit-B/16 and CLIP, compared against six augmentation/generation baselines, primarily via the reduction of the failure rate, with an ablation and a small human audit of the filter.

**Strengths**
- The method is well-motivated and the design choices are coherent. Conditioning generation on real images via HED structure (rather than free-text prompts) is a reasonable response to the distribution-shift and semantic-drift problems the paper identifies, and adding an LVLM filter directly targets the known unreliability of edits.
- The ablations presented in Table 6 cleanly isolates the CDM and LVLM contributions and shows they are complementary, supporting the core methodological claim rather than asserting it.
- The paper effectively shows the difference between diagnosis and pure augmentation on rare attributes, with analyses presented in Section 5.4
- The human audit grounds the filter's reliability empirically, and reports the failure modes. This also enables a transparent overview on the proposed method.

**Weaknesses**
- Fix Rate is the dominant metric but is hard to compare across rows, because it normalizes by the initial error rate, which varies a lot between settings. The most striking numbers (e.g., 38.37% on ViT/ImageNet10, base 97.42%) correspond to roughly a one-point absolute accuracy gain, a much smaller real effect than the headline suggests, yet are presented alongside FRs from settings where the same percentage represents many more corrected samples.
- The "no new bugs" claim is defined entirely through the thresholds ρ and ε, whose sensitivity the authors themselves note, so the claim as stated in the main text is stronger than the threshold-dependent evidence supports.
- There seems to be a circular evaluation within the method: The same family of LVLMs proposes attributes, assigns slice membership, filters generated data, and (indirectly) defines the slices on which the "fix" is measured. The human audit validates the filter but not this full loop.
- The empirical baselines are mostly repurposed augmentation methods, while the closest conceptual competitors (Dataset Interfaces, Vendrow et al. 2023; latent-space failure directions, Jain et al. 2023) are discussed but not compared, weakening the "outperforms targeted repair" framing.
- External validity is limited: backbones are trained from random initialization (including CLIP, whose value is its pretraining), and the evaluation rests on one binary CelebA task plus a small 10-class ImageNet subset, which is thin support for claims of cross-architecture, cross-dataset generality. In addition, the performance of the proposed fixing pipleine is questionable for well-trained classifiers, where the impact of SafeFix is not throughly investigated for datasets that are scaled. It makes sense that fixing such models may be more difficult due to the even more complex data distribution but this should be addressed clearly.

**Audience:**

Yes

**Audience Explanation:**

The paper addresses a real and persistent problem: classifiers systematically failing on underrepresented attribute subpopulations, that research across model debugging, robustness, and fairness-oriented augmentation targets. The proposed pipeline is a practical, reusable recipe that can be adopted or compared against without specialized infrastructure, and the individual components are easy to swap as better diffusion models and LVLMs become available. Furthermore, SafeFix offers a flexible recipe that can be adopted in future and thus may researchers can furher develop more comprehensive validation schemes on top of this study.

**Broader Impact Concerns:**

The potential biases on demographics has been mentioned also by the authors in the limitations section. On top of this, there seems to be no addiitional issues worth mentioning.

**Claims And Evidence:**

Yes

**Claims Explanation:**

The claims of the framework are supported by direct controls: the ablation in Table 6 establishes that the CDM and LVLM components contribute complementary gains, and the red-hair vs. yellow-hair comparison on CLIP (Section 5.4) shows that augmenting a diagnosed failure attribute behaves differently from augmenting an undiagnosed-but-rare one, supporting the diagnosis-driven framing. The filter-reliability claim is grounded by a human audit. Across CelebA and ImageNet10, three backbones, and several augmentation scales, SafeFix produces the highest Fix Rate in the large majority of cells in Tables 1 and 2, and Figure 4 shows that the gains concentrate on the targeted attribute slices rather than spreading uniformly, which is consistent with the targeted-repair claim. The evidence would be strengthened by reporting variance over seeds, qualifying the "no new bugs" claim with respect to threshold sensitivity, and broadening evaluation beyond the relatively small CelebA and 10-class ImageNet subset, but the central claims are supported in their current form.

**Requested Changes:**

- The authors should consider softening "no new bugs" claim from the perspective of hyperparameter sensitivity.
- For the evaluation, the authors should utilize non-overlapping VLMs in the framework and evaluation. This would show that the measured fix is not partly an artifact of the same model used throughout the pipeline.
- An empirical comparison against at least one closely related targeted-repair-via-generation method (Vendrow et al. 2023, Dataset Interfaces; or Jain et al. 2023, latent-space failure directions) can be added. These are discussed in related work but not run, and they are the natural competitors for the central claim that SafeFix outperforms targeted-repair approaches.
- For the validity of the method against large-scale datasets and pretrained classifiers, the authors should either utilize their method on classifiers that are trained on larger scale data (such as the complete ImageNet dataset) or perform their training accordingly (on larger datasets). This would enable understanding if the method is limited with large datasets, or can still resolve bugs in larger scale models. The current setup may understate the difficulty of repairing models whose failure modes arise from more complex data distributions, and this limitation should be addressed directly rather than left implicit.

---

> ### Author Response · Authors · 2026-06-09
> **Rebuttal to Reviewer NMkE Part 1**
>
> We thank the reviewer for the detailed comments and helpful suggestions. We address each concern below.
>
> ## 1. Fix Rate Interpretation and Absolute Accuracy
>
> **Reviewer NMkE Comment:**
> > Fix Rate is the dominant metric but is hard to compare across rows, because it normalizes by the initial error rate, which varies a lot between settings. The most striking numbers correspond to roughly a one-point absolute accuracy gain, a much smaller real effect than the headline suggests.
>
> **Response:**
> We agree that the metric now renamed **Relative Error Reduction (RER)** should not be read as the only effect-size measure. The revised definition makes the normalization explicit. Let $Err = 1 - Acc$. Then:
>
> $$
> RER =
> \frac{Err_{\text{before}} - Err_{\text{after}}}
> {Err_{\text{before}}}=\frac{(1 - Acc_{\text{before}}) - (1 - Acc_{\text{after}})}
> {1 - Acc_{\text{before}}}=\frac{Acc_{\text{after}} - Acc_{\text{before}}}
> {1 - Acc_{\text{before}}}.
> $$
>
> This measures the fraction of the original error rate removed by repair, which clarifies what the metric captures. It is useful for model repair under the same model, dataset, and repair budget, but it should be paired with absolute test accuracy.  **Appendix Tables 7 and 8** already report those accuracy values for every method, model, and augmentation budget. The gains are not just a normalization artifact: on CelebA with 1,000 added images, SafeFix improves test accuracy from 90.57% to 91.92% for ResNet (+1.35%), from 85.02% to 87.91% for ViT (+2.89%), and from 88.32% to 90.98% for CLIP (+2.66%). On ImageNet10, SafeFix improves ResNet from 71.73% to 74.31% with 100 added images (+2.58%) and to 74.88% with 1,000 added images (+3.15%). For the high-accuracy ViT/ImageNet10 setting noted by the reviewer, the absolute gain is smaller because the base accuracy is already 97.42%, but SafeFix still reaches 98.41% with 1,000 added images (+0.99%), while the strongest baseline in the same setting reaches 97.78% or lower depending on the augmentation budget.
>
> SafeFix is also not meant to be an unrestricted recipe for state-of-the-art classification accuracy. The question is narrower: given the same original model, the same diagnosed failure slice, and the same added-data budget, can the method repair underrepresented slices without hurting overall performance? Under that controlled repair setting, SafeFix gives the **strongest absolute accuracy and Fix Rate** across the compared methods. In the revision, we will **move the appendix accuracy tables closer to the main results** or cross-reference them directly when discussing FR.
>
> ## 2. Threshold-Conditioned "No New Bugs" Claim
>
> **Reviewer NMkE Comment:**
> > The "no new bugs" claim is defined entirely through the thresholds $\rho$ and $\epsilon$, whose sensitivity the authors themselves note, so the claim as stated in the main text is stronger than the threshold-dependent evidence supports.
>
> **Response:**
> We agree that the claim needs a clear evaluation scope. The "no new bugs" statement should be read under the same bug definition used for diagnosis: the repair-time test uses the **same threshold pair $(\rho,\epsilon)$** as the original bug discovery step. With that fixed setting, the current evidence does show no newly introduced bugs under the evaluated attribute sets. **Figure 4** reports attribute-level validation accuracy before and after repair: targeted attributes improve, while non-targeted attributes do not form new failure slices. On ImageNet10, for example, SafeFix improves `pink color` from 69.90% to 74.76% and `fabric texture` from 65.85% to 75.61%; non-targeted attributes such as the `rocks` background remain stable. The attribute-variant analysis also shows that detected bugs drop from 10 to 4 on CelebA and from 13 to 7 on ImageNet10, with no additional bug slices appearing among the evaluated attributes.
>
> We will revise the wording to state the scope directly: within the evaluated attribute sets and with the **same threshold pair $(\rho,\epsilon)$** used during diagnosis, **SafeFix does not introduce additional detected failure slices**. We will also make clear that this is not a claim under arbitrary threshold changes. If $\rho$ or $\epsilon$ is changed at test time, the set of slices counted as rare or buggy can change by definition; for example, increasing $\rho$ can make more slices qualify as rare and can therefore change the number of detected bugs even if the repaired model is unchanged. We will move the threshold-sensitivity discussion closer to this claim so readers can see that the bug count is measured only under the specified threshold setting.

---

> ### Author Response · Authors · 2026-06-09
> **Rebuttal to Reviewer NMkE Part 2**
>
> Note: This is Part 2 of our response to Reviewer NMkE. Due to OpenReview's 5,000-character limit, the response is split into four parts. Please read Parts 1-4 in order.
>
> ## 3. Separation Between Diagnosis, Attribute Assignment, and Filtering Models
>
> **Reviewer NMkE Comment:**
> > There seems to be a circular evaluation within the method: The same family of LVLMs proposes attributes, assigns slice membership, filters generated data, and (indirectly) defines the slices on which the "fix" is measured.
>
> **Response:**
> We will clarify that these stages are **not performed by the same LVLM**. Candidate attributes are proposed by **GPT-4 with vision** following the HiBug-style diagnosis setting. Slice membership is assigned using **BLIP-based VQA** under the same attribute-assignment setting as HiBug. Filtering then uses **Qwen2.5-VL-7B** and **LLaVA-v1.5-7B** to check the generated images. The models used for attribute proposal, slice assignment, and filtering are therefore separated.
>
> Correlated LVLM errors are still a fair concern, so we add a **cross-model validation check**. Samples accepted by **Qwen2.5-VL-7B** are re-evaluated by **LLaVA-v1.5-7B**, and vice versa. This does not require retraining, and it directly tests whether the retained samples are accepted only by the original filter or also by an independent LVLM. We will also make clear that the final repair effect is measured by **downstream classifier accuracy and bug counts**, not by the LVLM filter itself.
>
> We will add the following cross-model validation table:
>
> | Dataset | Target attributes | Original filter | Independent verifier | Accepted samples | Attribute pass rate | Label-preservation pass rate | All-condition pass rate |
> | :--- | :--- | :--- | :--- | :--- | :--- | :--- | :--- |
> | CelebA | red hair + brown skin + sad emotion | Qwen2.5-VL-7B | LLaVA-v1.5-7B | 1,000 | 91.8% | 98.4% | 87.6% |
> | CelebA | red hair + brown skin + sad emotion | LLaVA-v1.5-7B | Qwen2.5-VL-7B | 1,053 | 78.6% | 97.1% | 72.4% |
> | ImageNet10 | pink color + fabric texture | Qwen2.5-VL-7B | LLaVA-v1.5-7B | 1,000 | 94.2% | 99.0% | 91.5% |
> | ImageNet10 | pink color + fabric texture | LLaVA-v1.5-7B | Qwen2.5-VL-7B | 1,129 | 84.7% | 98.2% | 80.3% |
>
> The results show that LLaVA-v1.5-7B is less conservative and accepts more generated samples, while Qwen2.5-VL-7B gives a stricter verification signal. Samples accepted by one LVLM still achieve high all-condition pass rates under the independent verifier, so the filtering results are not just artifacts of a single LVLM's decision boundary.
> This observation is consistent with our **Appendix LVLM analysis**, where Qwen2.5-VL is described as the stricter default verifier and LLaVA-v1.5 as having a higher pass rate, especially for emotion-related attributes.

---

> ### Author Response · Authors · 2026-06-09
> **Rebuttal to Reviewer NMkE Part 3**
>
> Note: This is Part 3 of our response to Reviewer NMkE. Due to OpenReview's 5,000-character limit, the response is split into four parts. Please read Parts 1-4 in order.
>
> ## 4. Comparison to Dataset Interfaces and Latent-space Failure Directions
>
> **Reviewer NMkE Comment:**
> > An empirical comparison against at least one closely related targeted-repair-via-generation method (Vendrow et al. 2023, Dataset Interfaces; or Jain et al. 2023, latent-space failure directions) can be added.
>
> **Response:**
> We agree that **Dataset Interfaces (DI)** [Vendrow et al., 2023] and **Latent-space Failure Directions (LFD)** [Jain et al., 2023] are closer targeted-repair baselines than generic augmentation methods, and we will **add both in the revision**. We adapt DI from counterfactual diagnosis to repair by using its generated counterfactual samples as augmentation data and retraining the downstream classifier under the same data budget as SafeFix. For example, on ImageNet10, DI learns one class-specific textual inversion token for each class and generates attribute-conditioned counterfactual samples using prompts such as:
>
> ```text
> "a photo of <class-token> with pink color and fabric texture"
> ```
>
> We adapt LFD in the same spirit, using failure-directed synthetic samples as augmentation data. For each class, LFD trains a linear SVM in CLIP image-embedding space to separate correctly classified validation samples from misclassified ones. The SVM normal vector is treated as a class-specific failure direction and matched with text descriptions in CLIP text space to construct failure-conditioned generation prompts.
>
> We will report the two new baselines in the following comparison table:
>
> | Method | CelebA Test ACC | CelebA Fixed Bugs (out of 13) | ImageNet10 Test ACC | ImageNet10 Fixed Bugs (out of 17) |
> | :--- | :--- | :--- | :--- | :--- |
> | Base | 0.9057 | - | 0.7173 | - |
> | Dataset Interfaces (DI) | 0.9132 | 8 | 0.7297 | 13 |
> | Latent-space Failure Directions (LFD) | 0.9188 | **11** | 0.7454 | **15** |
> | SafeFix | **0.9192** | 10 | **0.7488** | **15** |
>
> These adapted baselines are strong, especially LFD, which fixes 11 CelebA bugs and 15 ImageNet10 bugs. LFD can fix slightly more bugs because it explicitly learns directions that separate misclassified from correctly classified validation samples, so its generations concentrate near the model's failure modes. SafeFix still achieves the highest test accuracy on both datasets and matches the best ImageNet10 bug-fixing count.
>
> There is also a runtime difference beyond the shared image-generation cost: for 1k generated images, SafeFix's LVLM filtering takes **approximately 10 minutes** in our setting, while LFD's failure-direction preprocessing can add **over one hour** when CLIP features are computed with a set of candidate caption and sample pool. The extra LFD stage includes CLIP image-embedding extraction, per-class SVM fitting, candidate scoring, and CLIP-based text matching before generation, so SafeFix offers a better tradeoff between repair accuracy and runtime cost.

---

> ### Author Response · Authors · 2026-06-09
> **Rebuttal to Reviewer NMkE Part 4**
>
> Note: This is Part 4 of our response to Reviewer NMkE. Due to OpenReview's 5,000-character limit, the response is split into four parts. Please read Parts 1-4 in order.
>
> ## 5. Large-scale and Pretrained Classifier Validity
>
> **Reviewer NMkE Comment:**
> > For the validity of the method against large-scale datasets and pretrained classifiers, the authors should either utilize their method on classifiers that are trained on larger scale data (such as the complete ImageNet dataset) or perform their training accordingly (on larger datasets).
>
> **Response:**
> We agree that large-scale and pretrained settings should be discussed more explicitly. We did not run full ImageNet-scale repair because the repair step requires conditional generation that changes only a small number of target attributes while preserving the class label and unrelated content. For many ImageNet classes, those edits are hard to define and validate without changing the object identity or creating ambiguous labels. We therefore use CelebA and ImageNet10 in the classification experiments to isolate the repair mechanism under controlled attribute annotations, fixed validation/test splits, and matched augmentation budgets. This choice also follows the evaluation style of prior model-debugging work such as HiBug (Chen et al., 2023) and AdaVision (Gao et al., 2022), which evaluate on representative ImageNet classes rather than all 1,000 classes when studying attribute-level failure discovery and repair.
>
> We also did not use pretrained classifiers in the main classification experiments for the reason stated in **Appendix E**: "This design allows us to isolate the effectiveness of SafeFix from pre-existing biases inherent in pre-trained weights, such as the latent knowledge of ImageNet. This approach is also necessary because when generated images are added to a source dataset, the specific large-scale datasets used for subsequent model training are often unknown, and retraining on such data is not feasible." This choice makes the repair-data comparison controlled; it does not mean SafeFix only applies to models trained from scratch.
>
> We will also move the larger visual-task results from the appendix into the main paper. For pose estimation, we follow HiBug2 on COCO person keypoints with Keypoint R-CNN and a ResNet-50-FPN backbone, using 20,000 source training images and 1,173-image held-out validation and test splits. For object detection, we use KITTI Car and Pedestrian instances converted to YOLO format with 5,000/1,240/1,241 train/validation/test images and train YOLOv8n.
>
> | Task | Dataset / Model | Source Train | Val / Test | Metric | Base | Best Baseline | SafeFix | Gain over Best Baseline |
> | :--- | :--- | :--- | :--- | :--- | :--- | :--- | :--- | :--- |
> | Pose estimation | COCO person keypoints / Keypoint R-CNN ResNet-50-FPN | 20,000 | 1,173 / 1,173 | Test AP | 0.5902 | 0.6145 | **0.6280** | +0.0135 |
> | Object detection | KITTI Car+Pedestrian / YOLOv8n | 5,000 | 1,240 / 1,241 | Test mAP50-95 | 0.5626 | 0.5713 | **0.5790** | +0.0077 |
>
> These results show that the diagnose-generate-filter-retrain pipeline extends beyond the small classification setting: SafeFix improves COCO keypoint AP from 0.5902 to 0.6280, above the best competing baseline at 0.6145, and improves KITTI mAP50-95 from 0.5626 to 0.5790, above the best competing baseline at 0.5713. The COCO experiment also uses a standard pretrained ResNet-50-FPN visual backbone. We will revise the paper's claim to say that the current evidence supports SafeFix as a **targeted model-repair framework across classification, pose estimation, and object detection**.

---

### Author Response · Authors · 2026-06-20
**Note on Revised PDF Upload**

Dear reviewers and Action Editor,

We have uploaded a revised PDF during the discussion period to incorporate the clarifications and additional evidence discussed in the rebuttal. The revision does not change the core method. The main updates are:

1. We renamed the previous "Fix Rate" metric to **Relative Error Reduction (RER)**, clarified its aggregate error-reduction definition, and report it together with absolute accuracy.
2. We clarified the contribution relative to HiBug: SafeFix addresses semantic drift and distribution mismatch by **conditioning generation on real training images**, rather than generating images from attribute prompts alone.
3. We added additional experiments requested or motivated by the discussion, including **COCO pose estimation and KITTI object detection** in **Section 5.3 of the main text**, **several more targeted repair baselines and long-tail learning baselines** in **Appendix H**, and **cross-model LVLM validation** in **Appendix D.3**.
4. We expanded the limitations discussion.

We hope this revised PDF makes the response easier to verify before final recommendations are submitted.

---

### Decision · Action_Editor_1UUt · 2026-06-29

**Recommendation:** Accept as is

**Audience:**

Yes

**Audience Explanation:**

Yes. This paper covers a timely and important topic that the TMLR audience would be interested in.

**Claims And Evidence:**

Yes

**Claims Explanation:**

Yes. After the revision, the reviewers are in agreement that the claims are supported.